# Directed evolution reveals the mechanism of HitRS signaling transduction in *Bacillus anthracis*

**Hualiang Pi**[1,2]**, Michelle L. Chu**[3]**, Samuel J. Ivan**[3]**, Casey J. Latario**[3]**, Allen M. Toth**[3]**, Sophia M. Carlin**[3]**, Gideon H. Hillebrand**[3]**, Hannah K. Lin**[3]**, Jared D. Reppart**[3]**, Devin L. Stauff**[3]**, Eric P. Skaar**[1,2]*

**1** Department of Pathology, Microbiology, & Immunology, Vanderbilt University Medical Center, Nashville, Tennessee, United States of America, **2** Vanderbilt Institute for Infection, Immunology, & Inflammation, Vanderbilt University Medical Center, Nashville, Tennessee, United States of America, **3** Department of Biology, Grove City College, Grove City, Pennsylvania, United States of America

* eric.skaar@vumc.org

**Data Availability Statement:** All relevant data are within the manuscript and its Supporting information files.

## Abstract

Two component systems (TCSs) are a primary mechanism of signal sensing and response in bacteria. Systematic characterization of an entire TCS could provide a mechanistic understanding of these important signal transduction systems. Here, genetic selections were employed to dissect the molecular basis of signal transduction by the HitRS system that detects cell envelope stress in the pathogen *Bacillus anthracis*. Numerous point mutations were isolated within HitRS, 17 of which were in a 50-residue HAMP domain. Mutational analysis revealed the importance of hydrophobic interactions within the HAMP domain and highlighted its essentiality in TCS signaling. In addition, these data defined residues critical for activities intrinsic to HitRS, uncovered specific interactions among individual domains and between the two signaling proteins, and revealed that phosphotransfer is the rate-limiting step for signal transduction. Furthermore, this study establishes the use of unbiased genetic selections to study TCS signaling and provides a comprehensive mechanistic understanding of an entire TCS.

## Author summary

Bacterial TCSs are a primary strategy for stress sensing and niche adaptation. Although individual domains and proteins of these systems have been extensively studied, systematic characterization of an entire TCS is rare. In this study, through unbiased genetic selections and rigorous biochemical analysis, we provide a detailed characterization and structure-function analysis of an entire TCS and extend our understanding of the molecular basis of signal transduction through TCSs. Moreover, this study provides a comprehensive map of point-mutations in these well-conserved signaling proteins, which will be broadly useful for studying other TCSs. The described genetic selection strategies are applicable to any TCS, providing a powerful tool for researchers interested in microbial signal transduction.

**Funding:** M.L.C, S.J.I, C.J.L, A.M.T, S.M.C, G.H.H, H.K.L, J.D.R and D.L.S. were supported by the Grove City College Swezey Fund and the Jewell, Moore, and MacKenzie Fund. E.P.S was supported by National Institutes of Health grant R01 AI73843. H.P. was supported by National Institutes of Health grant R01 AI73843 and National Institutes of Health 5T32HL094296. The funders had no role in study design, data collection and analysis, decision to publish, or preparation of the manuscript.

**Competing interests:** No authors have competing interests.

## Introduction

*B. anthracis* is a Gram-positive, spore-forming, facultative aerobe, and the causative agent of anthrax. *B. anthracis* spores can survive extreme temperatures, harsh chemical assaults, and nutrient-poor environments for many years [1]. This pathogen is one of the few infectious agents that have been proven effective as a weapon of bioterror. *B. anthracis* causes a variety of infectious syndromes including cutaneous, gastrointestinal, and inhalation anthrax. Inhalation anthrax occurs when *B. anthracis* spores enter a host through the respiratory system before disseminating to the lymph nodes and is the most deadly form of anthrax with a mortality rate approaching 90% [2]. To survive interactions with the host immune system during infection, *B. anthracis* has developed comprehensive systems for stress detection and detoxification [3]. Therefore, this pathogen is also an excellent model to study microbial stress responses.

Bacterial transcriptional changes in response to stress can be modulated by signal transduction systems known as two-component systems (TCSs). TCSs detect a wide range of signals and stressors including pH, temperature, nutrient, light, small molecules, envelope stress, osmotic pressure, and the redox state [4–10]. TCSs enable cells to sense, respond, and adapt to changes in their environment and regulate a wide variety of processes including virulence, sporulation, antibiotic resistance, nutrient uptake, quorum sensing, and membrane integrity [11–14]. A prototypical TCS consists of a membrane-bound sensor protein (histidine kinase, HK) and cytoplasmic response regulator (RR) [15–17]. A classic HK possesses five domains: a N-terminal Trans-Membrane domain (TM), a sensor domain, a HAMP domain that is commonly found in Histidine kinase, Adenylyl cyclases, Methyl-accepting chemotaxis protein, and Phosphatase, a DHp domain (Dimerization and Histidine phosphorylation), and a CA domain (Catalytic and ATP-binding) (Fig 1A) [15–19]. The latter two domains constitute the kinase core domains that harbor a number of well-conserved motifs (S1 Fig). A typical RR consists of two domains: a phosphorylation receiver domain and an output effector domain (Fig 1A, S2 Fig), with more than 60% of the latter being a DNA-binding domain [15–17]. In the presence of a specific stimulus, the HK detects the signal via the sensor domain, transmits the signal onto the DHp domain through the HAMP linker, phosphorylates its own conserved His located in the DHp domain, and then transfers the phosphoryl group onto a conserved Asp in the receiver domain of the cognate RR. In the case of the RR being a transcriptional regulator, this phosphorylation event activates the RR, induces homodimerization of the receiver domain, stimulates binding to the target promoters, regulates target gene expression, and modulates cellular physiology in response to environmental stimuli. TCSs are present in nearly all sequenced bacterial genomes as well as some fungal, archaeal, and plant species but are absent in animals and humans [15–17], making them attractive targets for antimicrobial therapeutics.

*B. anthracis* encodes approximately 45 TCSs, reflecting the complex environmental conditions encountered by this pathogen. A few *B. anthracis* TCSs have been studied [11, 12, 20], including a heme sensor system (HssRS) that responds to changes in available heme and activates the expression of a heme efflux pump upon heme exposure, and a HssRS interfacing TCS (HitRS) that activates an uncharacterized transporter (HitP). These two TCSs cross regulate at both HK-RR and post-RR signaling junctions, suggesting a link between heme toxicity and cell envelope stress [11, 12]. Although the nature of the activating signal of HitRS remains unclear, a high-throughput screen identified a series of cell-envelope acting compounds as inducers of HitRS [11, 21], including the small synthetic compound VU0120205 ('205) [11], nordihydroguaiaretic acid [22], chlorpromazine [23], targocil [24], and vancomycin [25]. These compounds share little structural similarity but each is implicated in cell envelope stress suggesting that HitRS detects perturbations in the cell envelope.

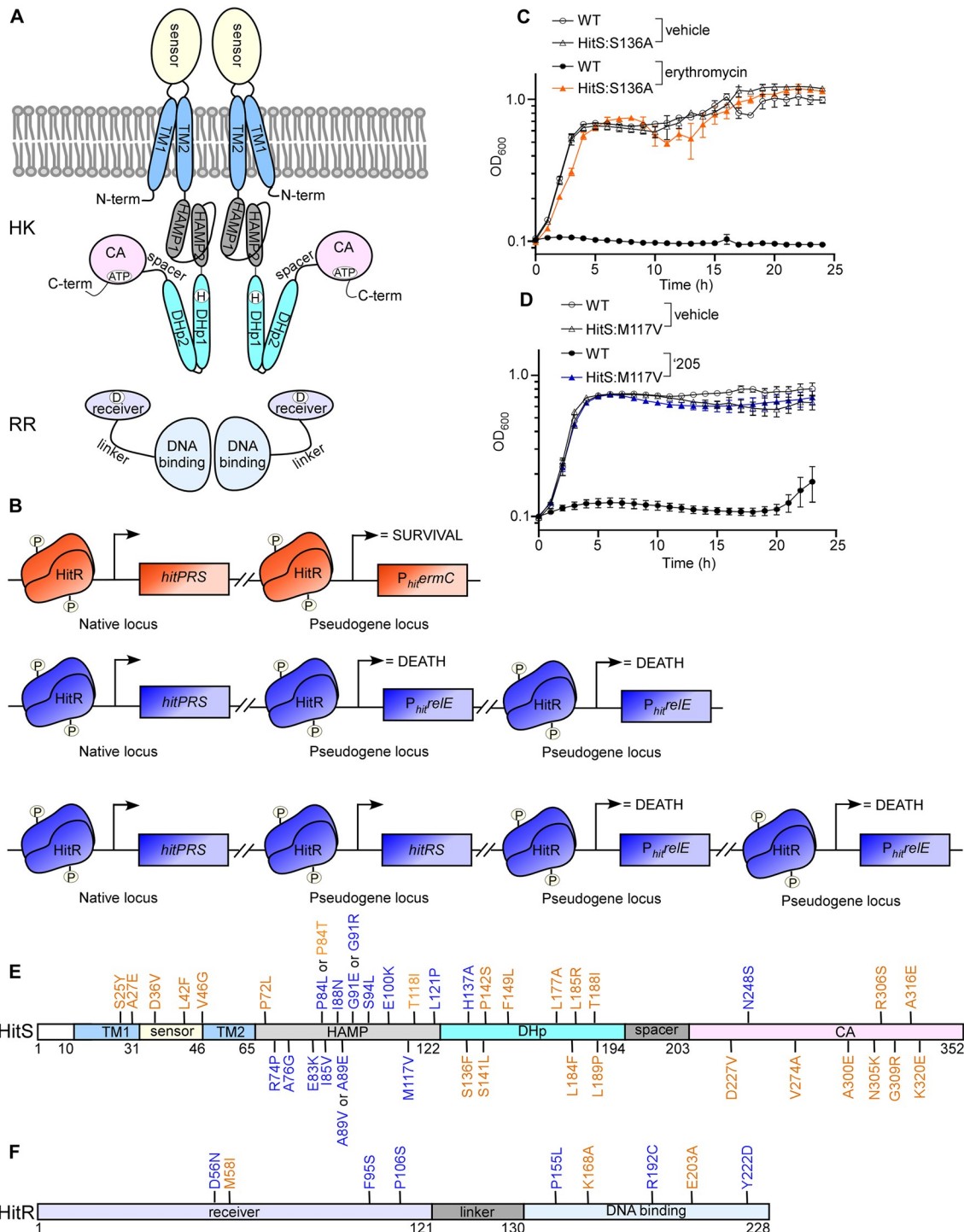

**Fig 1. Genetic selection strategies to study HitRS signaling mechanism.** (A) Schematics of the modular structure of a prototypical TCS. A classic histidine kinase (HK) consists of five domains: a N-terminal Trans-Membrane domain (TM), a sensor domain, a HAMP domain, a DHp domain, and a CA domain while a response regulator (RR) consists of two domains: a receiver domain and a DNA-binding domain. (B) Schematics for genetic selection strategies. To identify mutations that lead to constitutive activation of HitRS, an *ermC* strain shown in orange driven by a HitR promoter ($P_{hit}$) was used. The two strains shown in blue (*relE* strains) were employed to isolate inactivating mutations within the TCS genes. (C) Growth kinetics of *ermC* expressing strains (WT and a representative ON mutant HitS[S136A]) were monitored in the presence or absence of 20 μg ml⁻¹ of erythromycin. (D) Growth kinetics of *relE* expressing strains (WT and a representative OFF mutant HitS[M117V]) were monitored in the presence or absence of 20 μM '205. (E-F) Point mutations within HitS (E) and HitR (F) isolated from the genetic selections that lead to either inactivation (blue) or constitutive activation (orange) of this TCS.

In this study, genetic selection strategies were utilized to dissect the molecular mechanisms of signal transduction by HitRS. Numerous point mutations that lead to either inactivation or constitutive activation of the HitRS system were isolated. Representative point mutations were characterized biochemically to evaluate their effects on various activities required for signal transduction. These data uncovered the essential molecular determinants for HitRS stress sensing and signal transduction including: (i) four residues critical for the autokinase activity (S136/F149/V274/G309) besides the well conserved phosphoaccepting His and ATP-binding Asn, (ii) three residues essential for the phosphatase activity (S141/F149/R306), and (iii) five additional residues within HitR crucial for phosphotransfer and DNA-binding (F95/P106/P155/R192/Y222) besides the conserved phosphoaccepting Asp. In addition, our results revealed specific interactions among various domains and between HitR and HitS. Importantly, this study provides a detailed systematic characterization of TCS and expands our understanding of the molecular basis of signal transduction through TCS. Given that these signaling proteins are well conserved among distinct bacterial species, the described genetic selection and information obtained from this study may be broadly applicable across other TCSs.

## Results

### Devising genetic selections to study the mechanism of HitRS signaling

To dissect the molecular determinants within HitRS that are required for signal sensing and promoter activation, two sets of genetic selections were performed. To isolate mutations that lead to constitutive activation of HitRS, we created a *B. anthracis* strain harboring the erythromycin resistance gene *ermC* driven by the HitRS promoter ($P_{hit}ermC$) (Fig 1B). This strain was plated on medium containing toxic levels of erythromycin and colonies that arose represented bacteria that acquired mutations to constitutively activate the $P_{hit}$ promoter (Fig 1C). Thus we named this selection the "*ermC selection*" and the constitutive activating mutations "ON" mutations.

To identify critical residues within HitRS required for signal transduction, *B. anthracis* strains were created in which $P_{hit}$ drives expression of two copies of *Escherichia coli relE* ($P_{hit}relE$) (Fig 1B). The gene *relE* encodes an mRNA endoribonuclease that, when expressed following σ205-dependent activation of HitRS, cleaves mRNA leading to cell death. In addition, strains were created that harbor one or two copies of *hitRS*, the latter to select for strongly inhibitory variants of HitRS or mutations outside of *hitRS*. Colonies that arose from this selection represent strains containing mutations that render them unable to activate HitRS-dependent signal sensing and gene activation (Fig 1D). The employment of two copies of *relE* excluded mutations within *relE* and enabled preferential isolation of mutations within *hitRS* that inactivate signaling of this TCS. Therefore we named this selection the "*relE selection*" and inactivating mutations "OFF" mutations.

Three types of mutations were isolated from both selections: deletions, frame shifts, and point mutations. Point mutations enabled us to define residues critical for HitRS signal transduction and therefore were the focus of this study. Numerous point mutations were isolated that led to either inactivation or constitutive activation of HitRS, including 40 point mutations that were dispersed in different domains of HitS (2 in the TM domain, 3 in the sensor domain, 17 in the HAMP domain, 9 in the DHp domain, and 9 in the CA domain) (Fig 1E) and 8 point mutations in HitR (3 in the receiver domain and 5 in the DNA-binding domain) (Fig 1F). Among these point mutations, 28 were constitutively activating ON mutations while 20 were inactivating OFF mutations (Fig 1E and 1F). These point mutations enabled structure-function analysis to interrogate the roles of individual residues and domains and define the molecular basis of HitRS signaling.

## HitS is an intramembrane-sensing HK that detects cell envelope stress

HK sensor domains are highly variable, reflecting the wide variety of input signals that these proteins can sense. Signals perceived by the sensor domains are propagated to the cytoplasm through the TM helices. HitS contains two putative TM helices, the orientation and location of which were consistently predicted by multiple programs including SMART and TOPCONS [26, 27]. TM1 (residue 11 to 31) spans from cytosol to exterior while TM2 (residue 47 to 65) spans from exterior to cytosol. These two helices are connected by a 15-amino-acid sensor domain (Fig 1E). Notably, HKs with small sensor domains (≤25 amino acids) have been characterized as intramembrane-sensing HKs [28, 29]. This group of HKs detect signals within the membrane interface and are often involved in cell envelope stress [29], which coincides with HitRS being activated by several cell-envelope acting compounds [11].

Genetic selections identified five point mutations from this region: two (S25Y and A27E) in TM1 and three (D36V, L42F, and V46G) in the sensor domain. All of these mutations are ON mutations (Fig 1E), indicating that each mutation triggers a sufficient conformational change to enable HitS activation that normally only takes place upon stress detection or ligand binding. Several studies have shown that hydrophilic residues in the TM segment are important for signal recognition [18, 30, 31]. Indeed, S25 was substituted by a slightly polar Tyr while the hydrophobic A27 was substituted by a negatively charged Glu, indicating that it may be a common feature that hydrophilic residues of the TM helices participate in stress detection and signal transduction. There was no clear trend among the three mutations within the sensor domain but all led to constitutive activity of HitS in the absence of any inducers (Fig 1E). Very limited structural information is available to dissect the mechanism of ligand recognition and signal detection; however, this short sensor domain likely forms a small extracellular loop and binds ligand directly. Loop structures can accommodate diverse substitutions, which explains, at least in part, why the drastic changes from these substitutions are tolerated. Nevertheless, these data suggest that these residues identified from the genetic selections are important for ligand binding or stress sensing although the underlying mechanism remains to be elucidated.

## Essentiality of HAMP domain for HitS signaling

The input signal perceived by the sensor domain is subsequently transmitted to the intracellular signaling domains through transducing linkers such as the HAMP domain, which is found in approximately 30% of HKs [18]. HitS contains a putative HAMP domain immediately after TM2. To predict the tertiary structure of this domain, homology modeling of this segment (residue 66 to 121) was performed using I-TASSER with default settings [32]. This domain consists of two parallel helices (HAMP1 and HAMP2), connected by a flexible loop, and forms a homodimeric four-helical parallel bundle (Fig 2).

To understand the sequence properties and conservation pattern of the HAMP domain, a multiple sequence alignment was performed. HAMP domains are ~50 amino acids long and possess a small number of conserved residues including the Glu residue that marks the beginning of HAMP2 (Fig 2A). Interestingly, HitS contains two sterically restricted Pro residues in HAMP1. The first Pro (P72) is well conserved and its substitution to Leu converted HitS into a constitutively activating ON kinase while the second Pro (P84) is not conserved and its substitution yielded either a kinase-ON (P84T) or kinase-OFF (P84L) state (Fig 2). This signifies the importance of these Pro residues in HAMP function. All HAMP sequences adopt a heptad repeat pattern, in which positions *a* and *d* are occupied predominantly by hydrophobic residues (Fig 2A). These hydrophobic residues are critical for inter- and intramolecular interactions within the four-helix bundle [33]. Indeed, four hydrophobic residues in this region (I85, I88, M1117, and L121) were identified from genetic selections (Fig 2). I85 is equivalent to a

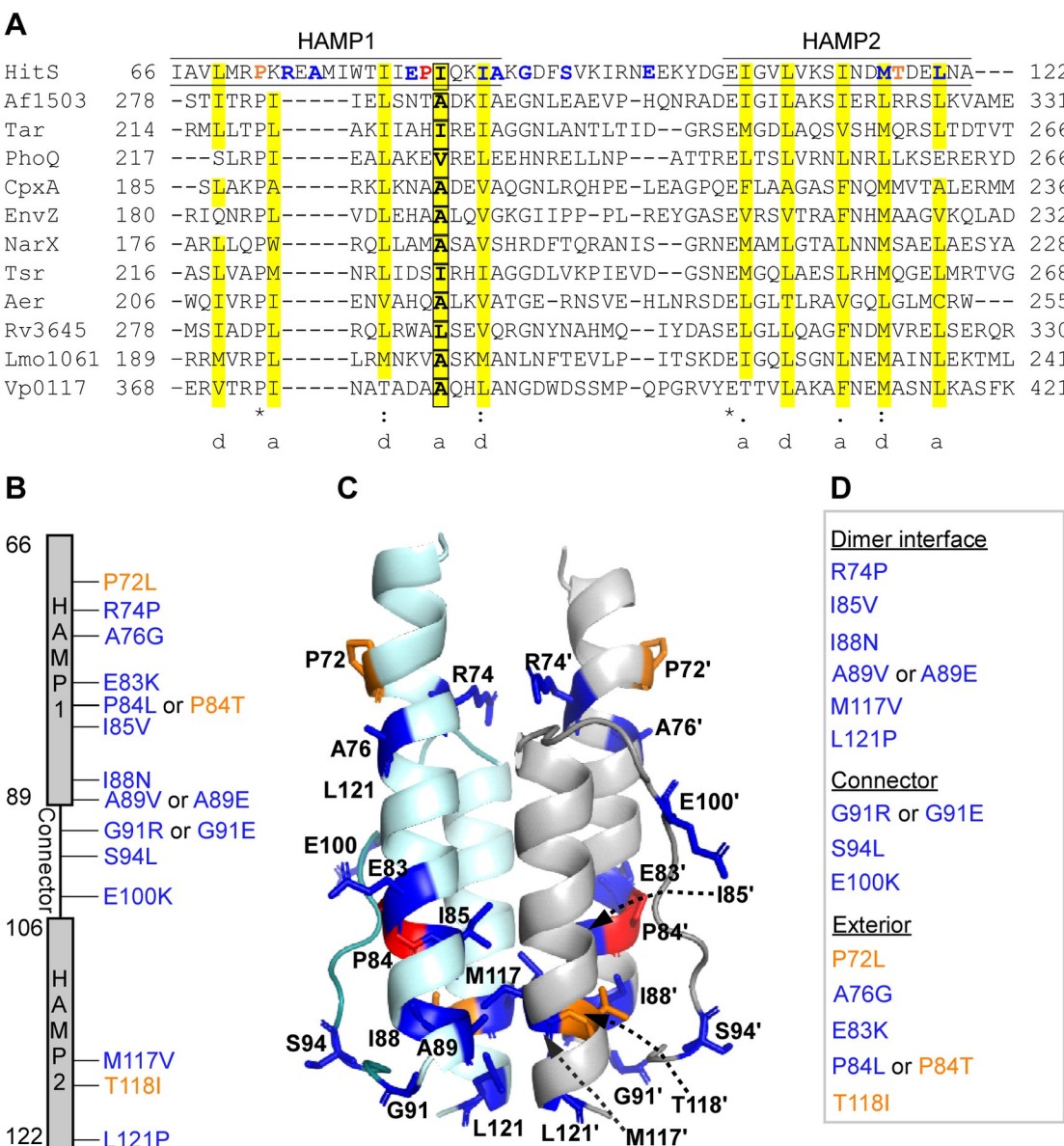

**Fig 2. Essentiality of HAMP domain for signal transduction.** (A) Multiple HAMP sequences were aligned to display the sequence property and conservation pattern of this domain. The two HAMP helices are underlined. The hydrophobic residues at the *a* and *d* positions of the heptad repeat pattern are highlighted in yellow and the residue in HAMP sequence that has been previously reported to be essential for signaling is shown in a box. The residues identified from genetic selections are highlighted in either orange or blue to specify either ON or OFF mutations, respectively. The residue Pro (P84) that can be mutated to either kinase ON or OFF state is highlighted in red. The sequences are from the following: HitS, *B. anthracis*; Af1503, *A. fulgidus*; Tar, *E. coli*; PhoQ, *E. coli*; CpxA, *E. coli*; EnvZ, *E. coli*; NarX, *E. coli*; Tsr, *E. coli*; Aer, *E. coli*; Rv3645, *Mycobacterium tuberculosis*; Lmo1061, *Listeria monocytogenes*; Vp0117, *Vibrio parahaemolyticus*. (B) All mutations isolated from genetic selections: OFF mutations are shown in blue while ON mutations are shown in orange. (C) All mutations are mapped onto the homology model based on *A. fulgidus* Af1503 (PDB ID: 4GN0). (D) All mutations were categorized into three groups based on their location in the homodimeric four-helix bundle.

residue that plays a critical role in signal transduction and that a substitution with a larger side-chain at this position promotes signaling while a smaller sidechain substitution compromises autokinase activity in other HKs [33]. In fact, substitution of I85 to valine with a slightly smaller sidechain completely suppressed the HitS response to stress and inactivated the HitRS system (Fig 2), demonstrating that even small changes at this position can drastically alter autokinase activity. Both M117 and L121 are located in HAMP2, and it seems likely that substitution of M117 to a bulky Val or L121 to a sterically restricted Pro may disrupt the α-helix conformation resulting in protein instability and loss of autokinase function. Therefore, we conclude that these hydrophobic residues are important for maintaining hydrophobic interactions within the four-helix bundle and are essential for HAMP function.

The flexible connector between the two HAMP helices plays an important role in stabilizing alternating conformations [34, 35]. It has been shown that a conserved glycine and two hydrophobic residues were the only residues critical for signaling function of a serine chemoreceptor [36]. Indeed, substitution of the Gly residue (G91) to either positively charged Arg or negatively charged Glu led to complete loss of autokinase function (Fig 2). Moreover, two additional residues were identified: S94 and E100. Substitution of the neutral S94 to the hydrophobic bulky Leu or mutation of the acidic E100 to positively charged Lys effectively inactivated the HitRS system (Fig 2), signifying the importance of this connector in TCS signaling through the HAMP domain.

It has been suggested that HAMP domains exist in two conformational states and the transition between the two alternating states is critical for signal transduction [15]. Therefore, mutations affecting interactions across the close interface within this dynamic four-helix bundle would result in a constitutive ON or OFF conformation, depending on the location of the residues [37, 38]. Consistent with this, we identified seven point mutations located in the dimerization interface: R74P, I85V, I88N, A89V or A89E, M117V, and L121P (Fig 2D), all of which resulted in a kinase-OFF state for HitS (Fig 2B). Collectively, the genetic selections identified 14 critical residues with 17 point mutations within the HAMP domain. The majority of these mutations are OFF mutations (14 out of 17; Fig 2), highlighting the essentiality of the HAMP domain to HitRS signal transduction.

## All selected point mutants of HitRS have potent growth phenotypes

To obtain a soluble form of HitS membrane protein, the N-terminal 67 amino acids were truncated and the intracellular region of HitS was cloned recombinantly (residue 68 to 352). Homology modeling of the truncated protein was performed using Phyre2 with default settings [39]. The homology model with the highest confidence covers the intracellular domain of HitS from residue 113 to 352. Genetic selections identified 22 point mutations in this region (S1E Fig), 11 of which were selected based on their locations on the structural model for further biochemical characterization: 2 in the HAMP domain, 3 in the DHp domain, and 6 in the CA domain. All 8 point mutations identified within HitR from the genetic selections were spread out in the two domains of HitR (Fig 1F) and subjected to further biochemical analysis to evaluate the effects of these mutation on activities required for signal transduction. Therefore, a total of 21 point mutants including the known phosphorylation-defective variants HitS$^{H137A}$ and HitR$^{D56N}$ [11] were selected for further study.

First the growth phenotype of these selected suppressor mutants was confirmed. As expected, the parental strain (WT P$_{hit}$ermC) for ermC selection showed no growth in the presence of 20 μg ml$^{-1}$ of erythromycin and reached a similar level of growth after a 15 h lag phase upon '205-mediated activation (S3 Fig), suggesting that accumulation of ermC expression is required for cells to gain resistance against this toxic level of erythromycin. All of the isolated

ON mutants showed potent resistance against erythromycin without significant growth delay. Addition of '205 provided no evident growth advantage to these mutants under these conditions (S3 Fig), suggesting that the erythromycin resistance gene is highly expressed in these constitutively activating ON mutants and inducers are no longer required to turn on the HitRS system.

For the parental strains used in the *relE* selection ($P_{hit}relE$), '205 was added to the medium to activate HitRS, leading to expression of *relE* and disruption of cell growth. Indeed, the *relE* strains grew very poorly with extended lag phases in the presence of '205: ~11 h lag phase for 2x(*relE*) or ~20 h for 2x(*relE* + *hitRS*) parent strain (S4 Fig). By contrast, '205 showed no inhibitory effects on any of the OFF mutants isolated and all OFF mutants grew remarkably well regardless of the presence or absence of '205 (S4 Fig). These results suggest that these OFF mutants were no longer responding to '205-mediated activation and their signaling activities were completely abolished.

To confirm the effects of these point mutations on transcription of the *hitPRS* operon, quantitative RT-PCR (qRT-PCR) was carried out. As expected, expression of *hitPRS* was upregulated in all ON mutants even in the absence of '205, with the HitS$^{S141L}$ mutation giving rise to the strongest activation for each gene (S5A Fig). Addition of '205 activates expression of *hitPRS* in the WT parental strain (WT $P_{hit}ermC$) but has negligible effects for all ON mutants, which explains why '205 provided little growth advantage to these mutants against erythromycin (S3 and S5B Figs). For some of the OFF mutants, the basal levels of *hitP* were notably lower compared to those in WT parental strains, particularly HitR$^{Y222D}$ and HitR$^{P155L}$ (S5C Fig). Addition of '205 induces expression of *hitPRS* in both WT parental *relE* strains, consistent with the poor growth phenotype with extended lag phases observed for these strains (S4 and S5D Figs). As expected, some OFF mutants showed no apparent response to '205 such as HitR$^{P106S}$, HitR$^{R192C}$, HitR$^{Y222D}$, and HitS$^{M117V}$. However, some OFF mutants exhibited a moderate response, particularly the mutants isolated from the *relE* strain carrying two copies of HitRS (2x(*relE* + *hitRS*)) including HitR$^{F69S}$, HitR$^{P155L}$, and HitS$^{N248S}$ (S5D Fig). All of these latter mutants were isolated from the ectopic copy of *hitRS*, indicating that these OFF mutations are dominant even though the chromosomal copy of *hitRS* was still intact and responsive to inducers in these mutants.

To further validate the results of the genetic selections, five representative point mutations were reconstructed in *B. anthracis* WT background and the effects of these chromosomal mutations on transcription of the *hitPRS* operon were evaluated using qRT-PCR. Expression of *hitPRS* was constitutively activated in all three ON mutants (HitR$^{M58I}$, HitS$^{T118I}$, and HitS$^{S141L}$) in the absence of the inducer while activation of *hitPRS* was completely abolished in both OFF mutants (HitR$^{R192C}$ and HitS$^{N248S}$) even in the presence of '205 (S5E and S5F Fig). These results demonstrate that the genetic selections are robust and powerful tools to dissect the molecular determinants that are crucial for HitRS signal transduction.

## Critical residues within HitRS stabilize the proteins and facilitate dimerization

All 21 point mutations were recreated using site-directed mutagenesis and mutant and WT proteins were purified. During the process of protein purification, we noticed some mutant proteins were unstable. Protein misfolding can lead to proteolytic degradation and subsequent protein inactivation. Indeed, four inactive OFF mutants were unstable: M117V and N248S in HitS, and P106S and Y222D in HitR, all of which showed apparent degradation upon SDS-PAGE (Fig 3A and 3B), indicating that these mutations led to defects in protein folding. Surprisingly, V274A, one of the HitS ON mutants, is partially unstable (Fig 3A). V274 marks

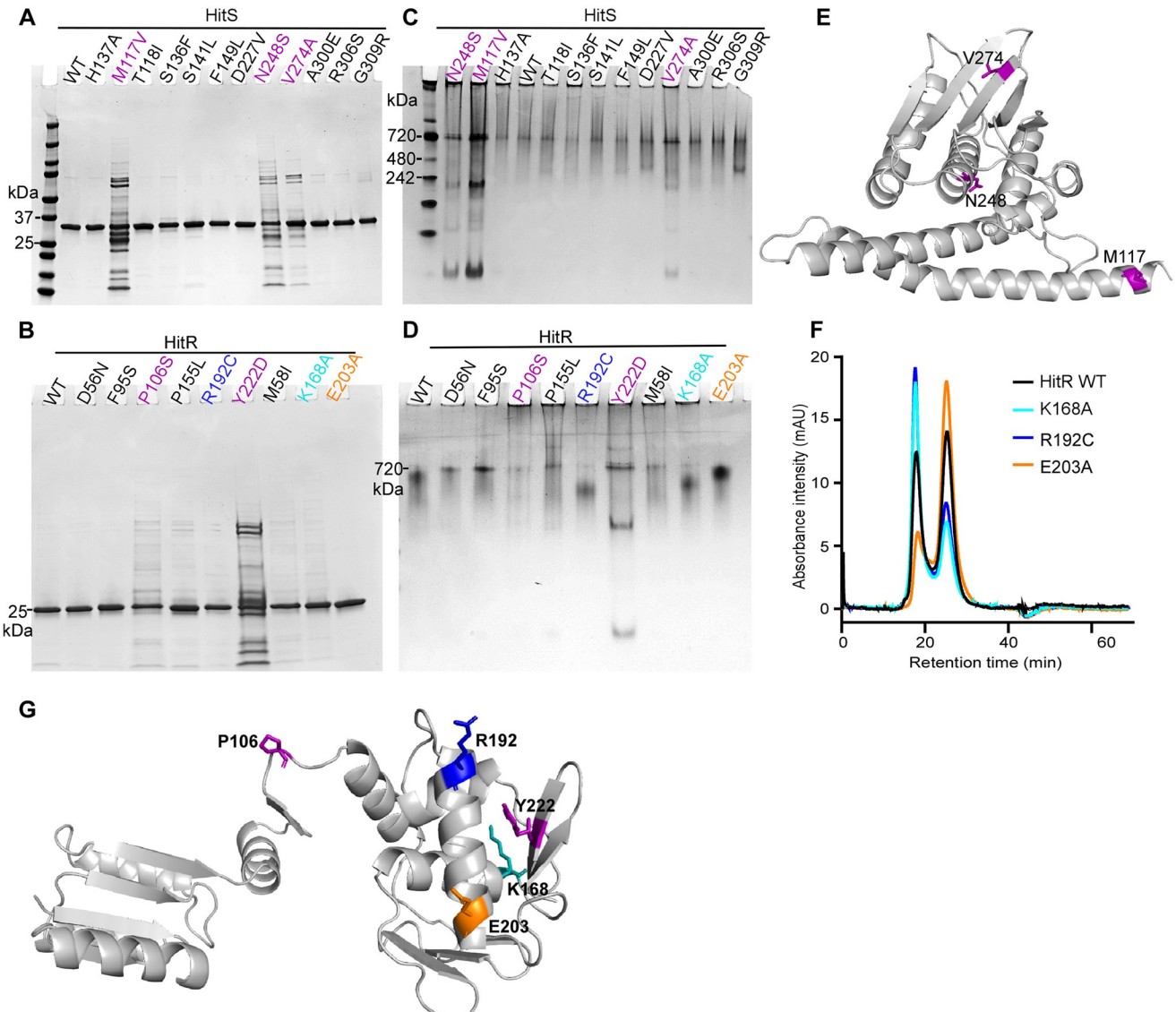

**Fig 3. Critical residues within HitRS stabilize protein and facilitate dimerization.** To evaluate the effects of HitR and HitS mutations on protein stability and dimerization, WT and mutant proteins were loaded onto SDS-PAGE (A, B) or native gels (C, D). Results shown are WT proteins and all mutants of HitS (A, C) or HitR (B, D) selected for further biochemical characterization. (F) Three HitR mutants affected protein dimerization as determined by size exclusion chromatography. Mutations that affect either protein stability or dimerization are mapped onto the homology models of HitS (E) or HitR (G). HitR model was generated based on *M. tuberculosis* RegX3 (PDB ID: 2OQR).

the beginning of the D-box and is located in one of the antiparallel β-sheets that hang over the ATP-binding pocket (Fig 3E and S1C Fig). The C-beta branched Val residue is bulky and suitable for β-strand conformation compared to Ala with a small sidechain. Thus, this substitution might disrupt the β-strand conformation resulting in defects in protein folding. It is intriguing how this V274A ON mutation promotes phosphorylation at the expense of protein stability.

Most HKs and RRs have been demonstrated to function as homodimers [15, 18]. The dimer interface is located in the DHp domain of the HK while the receiver domain is dimerized upon RR activation triggered by phosphorylation. To determine the effects of the point mutations on dimerization status, WT and mutant proteins were subjected to non-denaturing

native gel electrophoresis to analyze their mobility patterns in the folded state. No significant differences were observed for all HitS mutants compared to WT except that proteolytic degradation of unstable mutants was apparent in the native gel (Fig 3C), consistent with the prior results (Fig 3A). All the HitR mutants in the receiver domain formed a relatively sharp band similar to the mutant D56N, which is known to be an inactivating mutant due to loss of phosphotransfer capability [11]. Surprisingly, all four point mutants in the DNA-binding domain (K168A, R192C, E203A, and Y222D) migrated differently through the gel compared to WT (Fig 3D). Multiple variables could contribute to differences in migration including charge-to-mass ratio, folding status, and physical shape of the protein, which makes it challenging to interpret the different patterns observed. To further examine the dimerization status of these mutants, HitR WT and all these mutants except the unstable Y222D mutant were subjected to size exclusion chromatography, which separates proteins on the basis of molecular weight. The data showed that the composition of WT HitR was about 45% dimer and 55% monomer in solution (Fig 3F). As expected, the ON mutation K168A promoted dimerization and drove the equilibrium towards the dimeric form with an increase of 25% (Fig 3F), indicating mutation of this polar residue (K168) to a slightly hydrophobic Ala apparently facilitates hydrophobic interactions between monomers. However, it was surprising to note that the OFF mutant R192C promoted dimerization while the ON mutant E203A disrupted dimerization *in vitro* (Fig 3F). This seemed counterintuitive, however some RRs have been shown to form two types of dimers in distinct orientations and only the dimer in the correct orientation is active [40], which could explain this contradictory observation. In addition, all three substitutions followed the same trend of replacing hydrophilic residues (K168, R192, or E203) with hydrophobic residues (Ala or Cys), indicating that introducing hydrophobic residues at these positions (Fig 3G), particularly K168 and R192, enhances hydrophobic interactions and facilitates dimerization. An important caveat is that RR dimerizes upon phosphorylation and the results may not reflect the dimerization status of these mutants during signal transduction *in vivo*. Thus a thorough evaluation is required to dissect the effects of these mutations on other activities such as phosphotransfer and DNA-binding. Nonetheless, these data suggest that residues in the DNA-binding domain interact with the dimer interface of the receiver domain and may affect TCS signaling through modulating HitR dimerization status.

## The autokinase activity of HitS can be modulated in four different manners

To understand how one single-residue mutation alters protein function and locks a protein in a constitutively on or off state, we tested the effects of mutations on different activities intrinsic to the protein including the autokinase activity. Consistent with a prior study [11], substitution of the well-conserved phosphoaccepting H137 to Ala abolishes the autokinase activity (Fig 4A and 4B). Likewise, the two OFF mutations (M117V and N248S) led to a complete loss of autokinase activity (Fig 4A and 4B). The M117V mutation inactivates autokinase activity potentially by disrupting the α-helix conformation of the HAMP domain while N248S does so likely by disrupting the hydrogen bonds between Asn and ATP adenine resulting in abolished ATP-binding. In addition, two of the ON mutations (D227V and R306S) promoted autokinase activity and several ON mutants showed similar autokinase activity compared to WT (Fig 4A and 4B). However, we were surprised to note that four ON mutants exhibited significantly reduced autokinase activity: ~10-fold reduction for S136F, ~3-fold reduction for F149L, ~2-fold reduction for V274A, and ~4-fold reduction for G309R relative to WT. To better understand how these ON mutations affect autokinase activity, autophosphorylation kinetics of HitS WT and ON mutants were monitored for 15 min. Three distinct groupings were revealed: (i) some mutations facilitated the autokinase activity with a higher kinetic rate

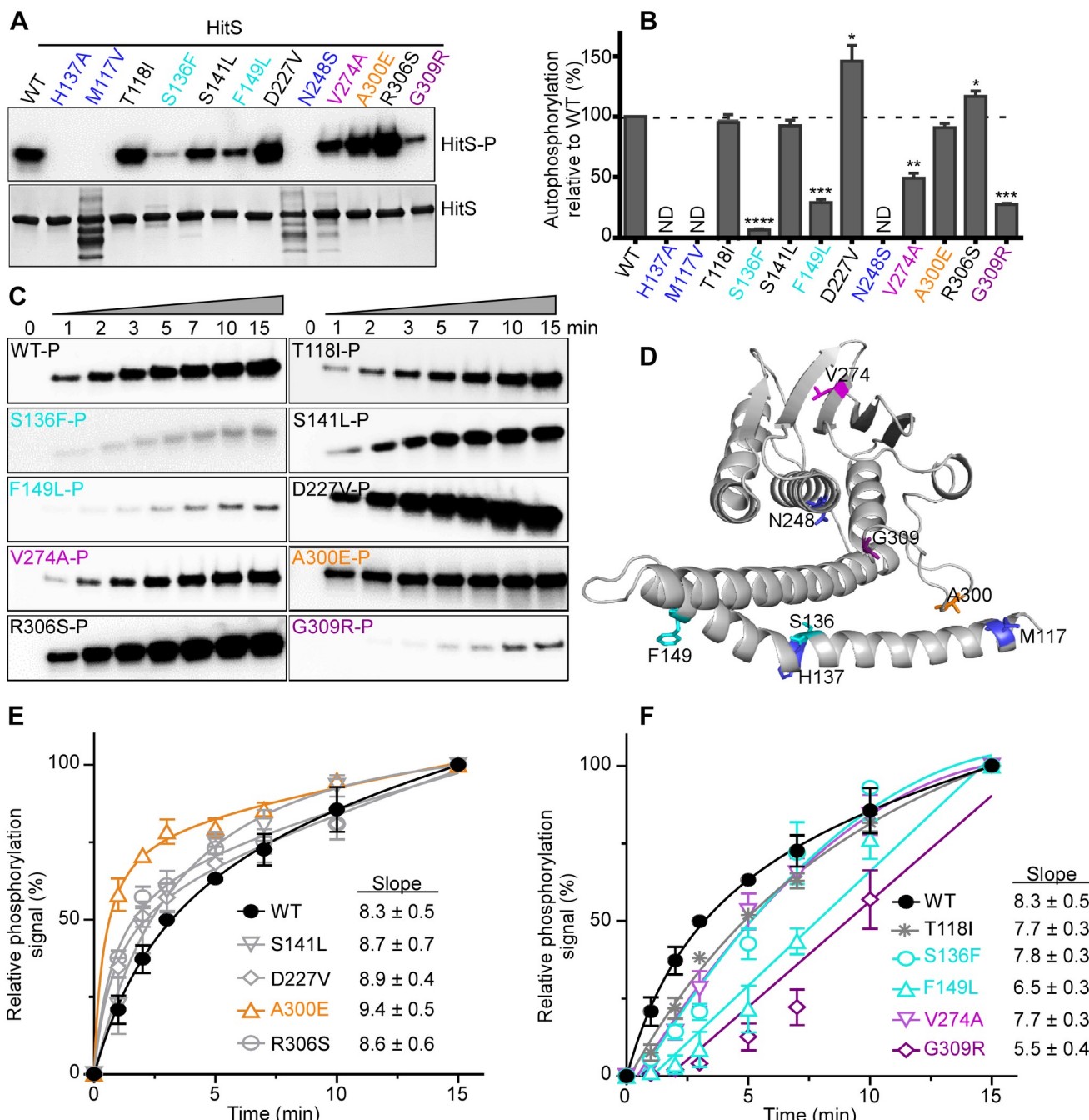

**Fig 4. The autokinase activity of HitS can be modulated in four different manners.** To evaluate the effects of mutations on HitS autokinase activity, the autophosphorylation efficiency of HitS WT and mutants was investigated. (A) Representative phosphor-images (top panel) to show autophosphorylation of HitS WT and mutants that were incubated with ATP [γ-$^{32}$P] for 30 min and quantified using a phosphoimager. The bottom panel is to show the amount of protein used for each reaction in an SDS-PAGE gel. The intensity of the phosphorylation signal was quantified and four independent experiments are shown in (B) (mean ± SEM). Significant differences between WT and each mutant are determined by two-tailed t-test, where $^{*}P < 0.05$, $^{**}P < 0.01$, $^{***}P < 0.001$, and $^{****}P < 0.0001$. (C) Representative phosphor-images to show the kinetics of autophosphorylation by HitS WT or ON mutants, which was monitored for 15 min. (D) Mutations that affect autophosphorylation are mapped onto the HitS model. (E, F) For better visualization, the intensity of the phosphor-signal at different timepoints was quantified and three independent experiments are presented in (E, F) (mean ± SEM). Mutants were organized into two graphs based on their autokinase activity. Data of the first four timepoints (i.e., 0, 1, 2, and 3 min) in E and F were used for slope determination by linear regression analysis.

(A300E, D227V and R306S), (i) some mutations showed minor effects (S141L and T118I), and (iii) some mutations disrupted the autokinase activity with a lower rate resulting in significantly diminished autokinase yield (S136F, F149L, V274A, and G309R) (Fig 4C–4F and S3 Table). Thus, we conclude HitS autophosphorylation can be modulated in four different manners including abolished activity observed in the OFF mutants (Fig 4). Furthermore, these data uncovered four additional residues critical for the autokinase activity: S136/F149 adjacent to the phosphorylation site and V274/G309 from the CA domain, besides the well conserved phosphoaccepting His and ATP-binding Asn. However, the observation that the ON mutants exhibited reduced autokinase activity appeared contradictory, suggesting that the phosphotransfer rates of these mutants might be altered to compensate for this reduction.

## Phosphotransfer is the rate-limiting step for signal transduction

Next we examined the impact of all HitS ON mutations on phosphotransfer efficiency. HitS WT or each ON mutant was autophosphorylated with $[\gamma\text{-}^{32}P]$-ATP and phosphotransfer from HitS WT or mutant to HitR WT was then monitored for 15 min. Indeed, all ON mutants transferred the phosphorylation signal significantly faster than WT, including the mutants with defects in autokinase activity such as S136F, V274A, and G309R (Fig 5). This indicates that these mutations are functional trade-offs where the autokinase activity is compensated, at least in part, by faster phosphotransfer. In addition, it is important to note that the phosphotransfer took place in an instantaneous manner. More than 60% of the phosphor signal was transferred from HitS to HitR within 15 seconds (Fig 5A and 5C). Thus we conclude that phosphotransfer from HitS to its cognate regulator HitR is the rate-limiting step for signal transduction.

## Critical residues required for the phosphatase activity of HitS

Many HKs are bifunctional enzymes that function as both kinases and phosphatases [41]. The autokinase-competent and phosphatase-competent states need to be maintained in balance and modulated in response to specific environmental cues. Importantly, dephosphorylation is not a simple reverse reaction of phosphorylation and may require different residues to achieve this activity. To define the crucial residues required for HitS phosphatase activity, we examined the effects of HitS mutations on dephosphorylation of its cognate regulator HitR. Briefly, the GST-PmrBc fusion protein [42] was autophosphorylated and served as a phosphor donor, and the phosphoryl group was subsequently transferred to HitR WT protein. Dephosphorylation of the resultant phosphorylated HitR WT protein was then monitored for 60 min. First, we evaluated the three OFF mutants. H137A abolished the autokinase activity completely (Fig 4A and 4B); however, the phosphatase activity was intact and comparable to that of WT (Fig 6A and 6C), suggesting that the phosphoaccepting His is dispensable for the phosphatase activity and HitS is therefore not a reverse phosphatase. The two other OFF mutants (M117V and N248S) showed significantly compromised phosphatase activity (Fig 6A and 6C) with abolished autokinase activity (Fig 4A and 4B), likely due to instability of these two mutants (Fig 3A). RR dephosphorylation can be catalyzed by either HK-mediated dephosphorylation or auto hydrolysis. The latter is probably why minimal dephosphorylation activity was still observed. Next, we tested the four ON mutants located in the DHp domain, two of which showed drastically diminished activity in dephosphorylation including S141L and F149L (Fig 6A and 6D). We then examined the five ON mutants located in the CA domain, only one (R306S) of which exhibited reduced activity in dephosphorylation (Fig 6A and 6E). When the phosphatase activity of HK is disrupted, this eliminates its ability to remove the phosphoryl group from its cognate RR and the phosphorylated RR can stay active longer thereby

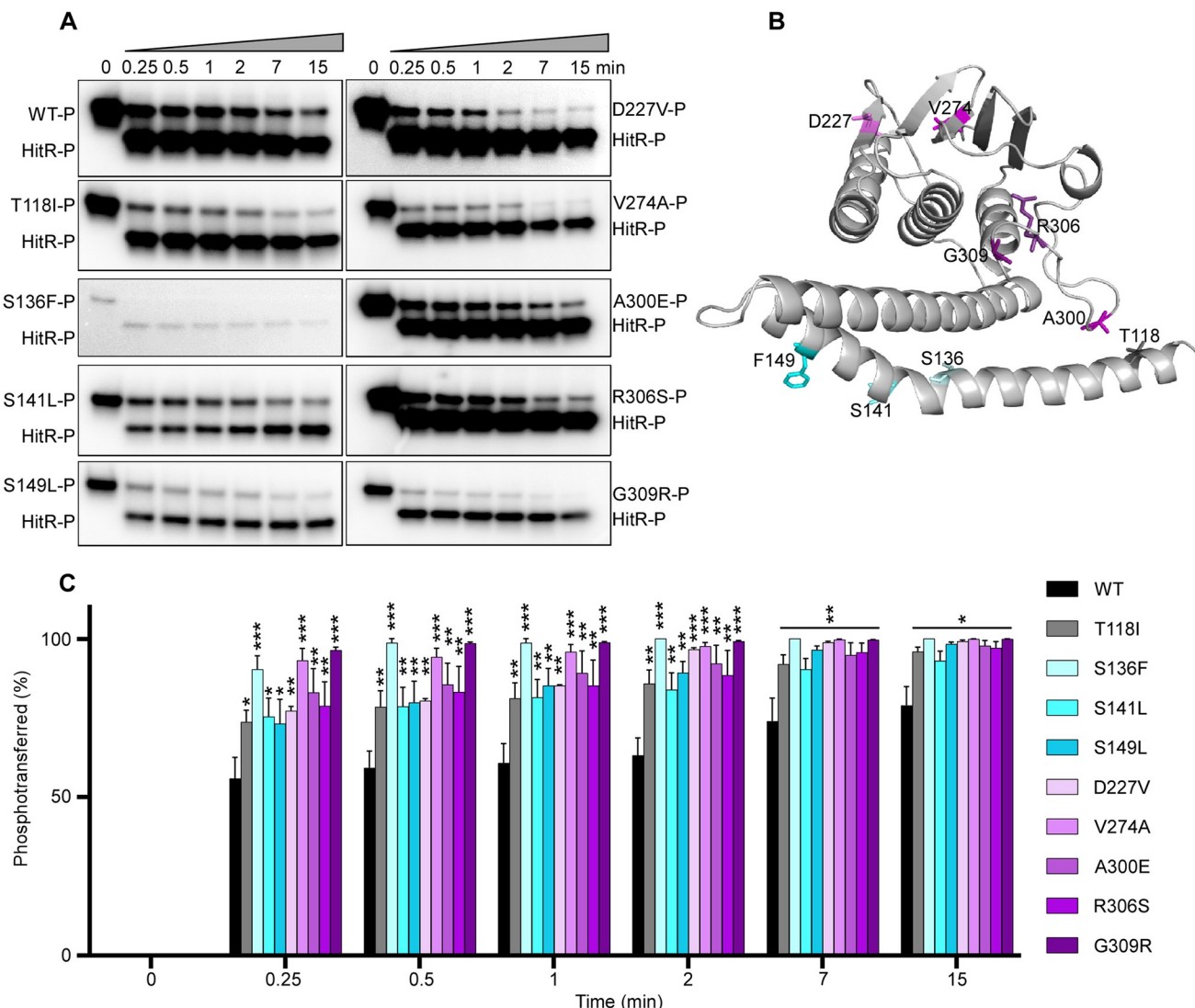

**Fig 5. Phosphotransfer is the rate limiting step for signal transduction.** To evaluate the effects of HitS ON mutations on transferring the phosphorylation signal, phosphotransfer efficiency of HitS WT or ON mutants to HitR WT was examined. (A) Representative phosphor images to show the kinetics of phosphotransfer from HitS WT or ON mutants to HitR WT, which was monitored for 15 min. (B) Mutations tested are mapped onto the HitS model. (C) The intensity of the lower band (signal transferred) was quantified and relative phospho-signal transferred at different time-points was calculated. Data shown in (C) are from three independent replicates (mean ± SEM). Significant differences determined by two-tailed t-test were observed between WT and each individual activating mutant, where $^*P < 0.05$, $^{**}P < 0.01$, and $^{***}P < 0.001$.

promoting signal transduction and gene activation. We conclude these three residues (S141, F149, and R306) are critical for phosphatase activity. Furthermore, these data demonstrated that both HAMP and DHp domains are important for dephosphorylation and not all residues critical for autokinase activity are required for dephosphorylation.

## Residues essential for HitR activation and specific interactions within HitR

RRs function as phosphorylation-triggered switches that mediate cellular physiology in response to environmental cues largely through two steps: phosphotransfer from HK to RR

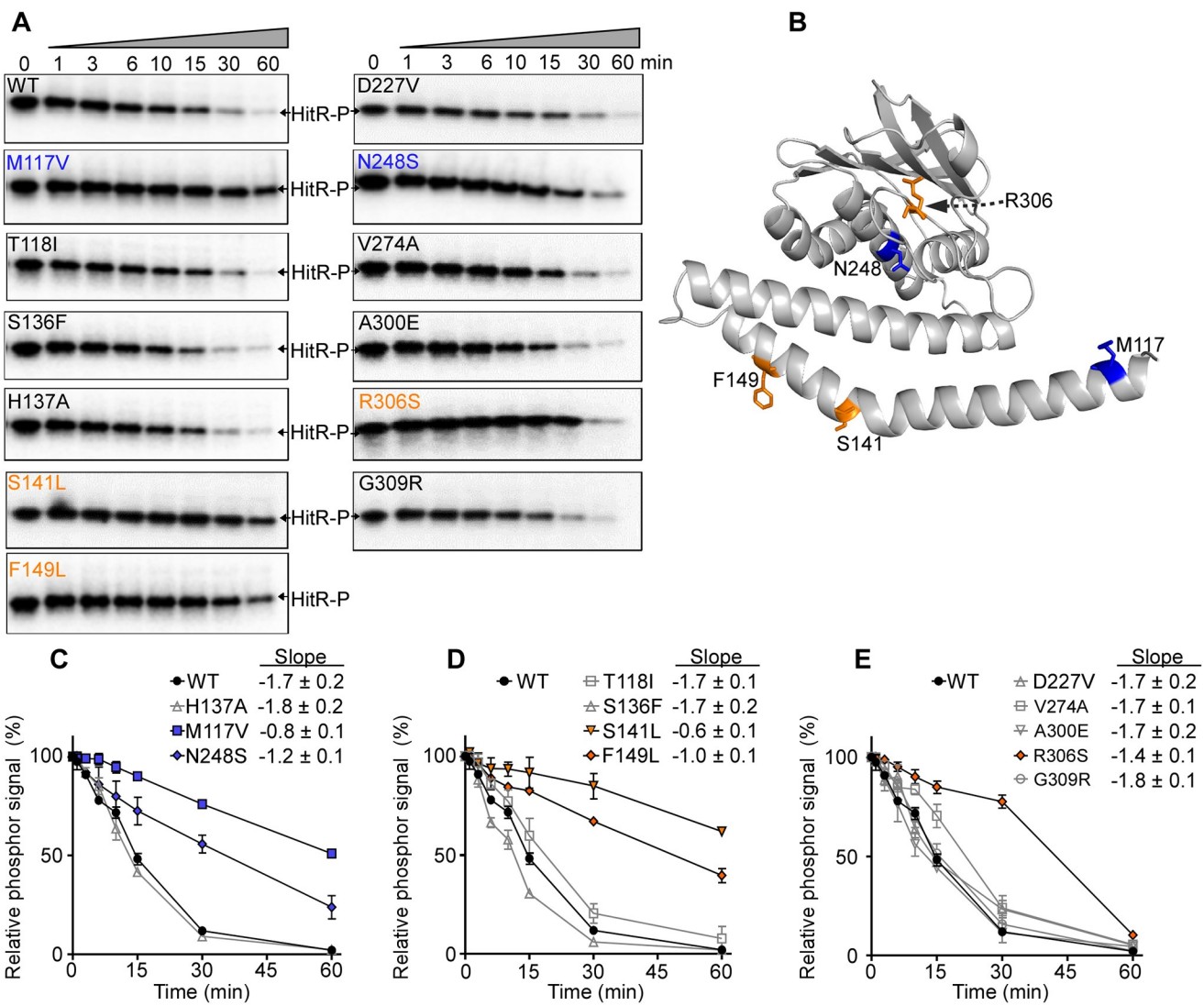

**Fig 6. Critical residues required for the phosphatase activity of HitS.** To evaluate the effects of HitS mutation on its phosphatase activity, dephosphorylation efficiency of HitS WT and mutants was tested. (A) Representative phospho-images showing the dephosphorylation kinetics of HitS WT or mutants using phosphorylated HitR WT, which was monitored for 60 min. (B) Mutants with altered phosphatase activity were mapped onto the HitS model. (C-E) The intensity of phospho-signal was quantified and relative phospho-signal remaining at different time-points was calculated. Data shown are three independent replicates (mean ± SEM). Mutants were organized into three graphs for optimal visualization. Data of the first four timepoints (i.e., 0, 1, 3, 6 min) for each protein were used for slope determination by linear regression analysis.

and RR-DNA-binding. To examine the effects of HitR mutations on signal reception, the phosphotransfer efficiency was evaluated in HitR WT and HitR mutants. HitS WT was autophosphorylated with ATP [γ-$^{32}$P] and the phosphoryl group was subsequently transferred to HitR WT or mutant proteins. Consistent with a prior study [11], substitution of the conserved phosphoaccepting Asp (D56) to Asn abolishes phosphor signal reception (Fig 7A and 7B). Two of the three ON mutations (M58I and K168A) showed significantly enhanced activity in phosphotransfer (Fig 7A and 7B). Among five OFF mutations, only P106S showed ~50% reduction of activity in phosphotransfer compared to WT (Fig 7A and 7B), which could be explained by protein instability (Fig 3B). However, it was intriguing that Y222D mutation

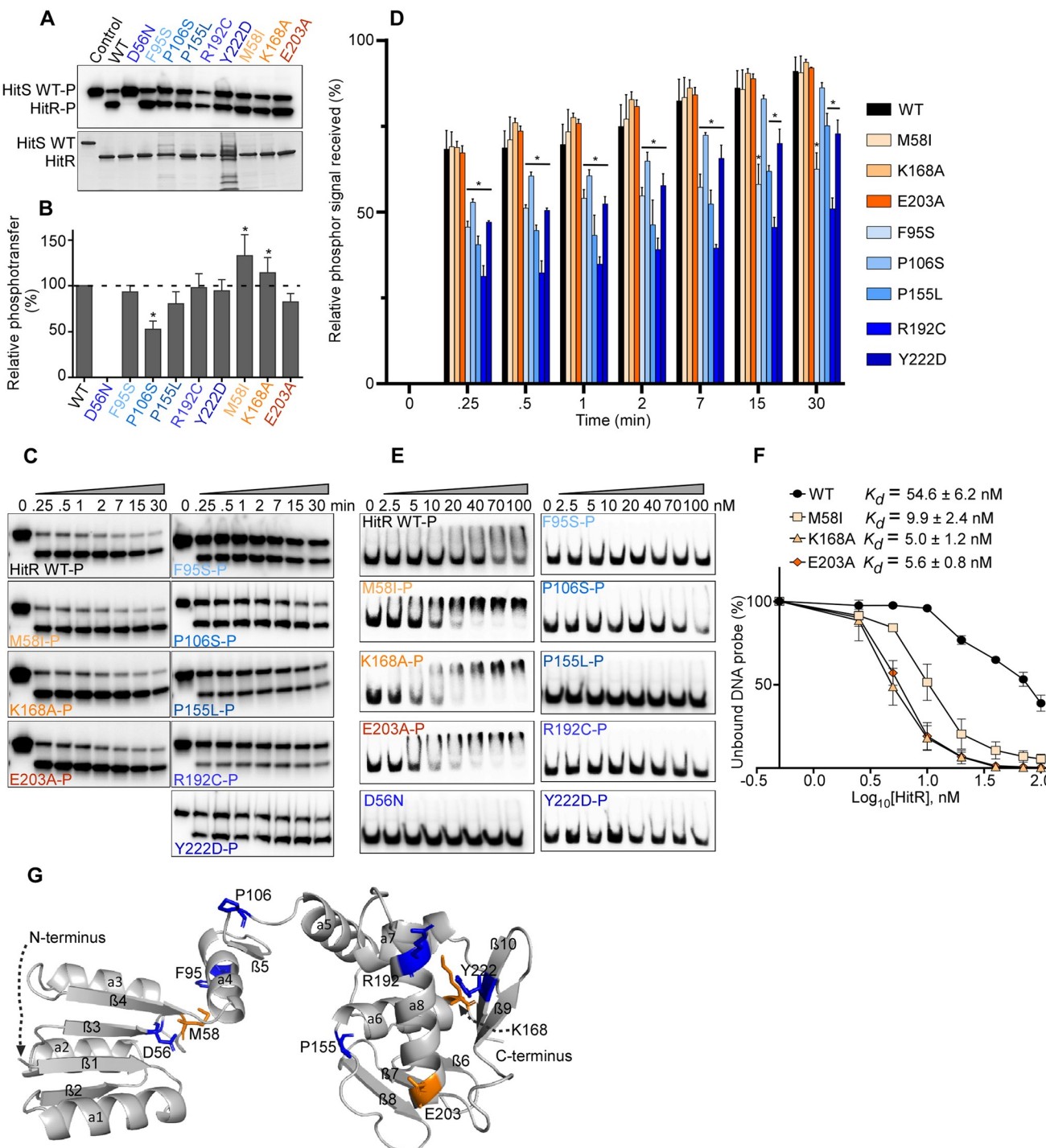

**Fig 7. Residues essential for HitR activation and specific interaction within HitR.** To examine the effects of HitR mutations on signal reception and DNA-binding, the phosphotransfer efficiency and DNA-binding affinity of HitR WT or mutants were tested. (A, B) Phosphotransfer efficiency from HitS WT to HitR WT or mutants was quantified. (A) The top is a representative phosphor image and the bottom is an SDS-PAGE gel showing the amount of protein used for each reaction. The intensity of radioactive signal was quantified and averages from four independent experiments are shown in (B) (mean ± SEM). Statistical significance was determined by two-tailed $t$-test, where $^*P < 0.05$. (C) Kinetics of phosphotransfer from HitS WT to HitR WT or mutants was monitored for 30 min. Representative images are shown. The intensity of the lower band (phosphotransferred) at each timepoint was quantified. Presented are averages from three independent replicates (D) (mean ± SEM). (E) Representative images to show DNA-binding of HitR WT or mutants to its target promoter evaluated by EMSA. (F) The band intensity of unshifted DNA probe (lower band) was quantified using GelQuantNET. All data points from three independent experiments were plotted and subjected to $K_d$ determination using GraphPad Prism 8 (mean ± SEM). (G) All point mutants tested are mapped onto a HitR structure model. All blue colors indicate OFF mutation while orange colors indicate ON mutation.

located in the DNA-binding domain exhibited an equivalent level of phosphotransfer activity compared to WT in spite of being the most unstable HitR mutant (Figs 7A and 7B and 3B).

To better understand how these mutations affect phosphor signal reception, the kinetics of phosphotransfer from HitS WT to HitR WT or mutants was monitored for 30 min. None of the ON mutants exhibited an accelerated rate of phosphotransfer while all OFF mutants displayed significantly slower kinetic rates relative to WT HitR (Fig 7C and 7D). M58 is in very close proximity to the phosphoaccepting residue D56 (Fig 7G) and the substitution to Ile appeared to facilitate phosphotransfer, indicating that M58 is involved in phosphorylation signal reception. K168 marks the C-terminal end of α6 in the DNA-binding domain and its substitution to Ala promoted phosphotransfer (Fig 7A and 7B), suggesting that the DNA-binding domain interacts with the receiver domain and facilitates phosphorylation reception. The data also indicate that α6 helix may be part of the interface of these two domains and important for their interaction between these two domains. P155 marks the end of the flexible loop between β8 and α6 in the DNA-binding domain (Fig 7G), and the tight turn created by P155 was likely destroyed by Ser substitution resulting in defects in phosphotransfer (Fig 7A–7D). This indicates that this loop not only enables conformational changes transmitted from the receiver domain to the DNA-binding domain, but also facilitates phosphotransfer from HK to RR. Both R192 and Y222 are located in the DNA-binding domain. R192 resides in helix α8 and Y222 is in the last β-strand. Interestingly, mutation of either residue affected reception of the phosphoryl group (7C-D), exemplifying the importance of the DNA-binding domain during phosphotransfer and interaction between the two domains of RR.

Next, we investigated the effects of HitR mutations on HitR-DNA-binding ability using an electrophoretic mobility shift assay (EMSA). HitR WT or mutant protein was first activated by phosphorylation, and the binding affinity to the target promoter was then evaluated. HitR WT binds DNA with an affinity of ~55 nM, and as expected, the D56N mutation led to a complete loss in DNA-binding. Remarkably, no visible band-shift was observed for all five OFF mutations with up to 100 nM of protein tested except P106S, which showed a smear with the highest protein concentrations (70–100 nM), indicative of a protein-DNA complex (Fig 7E). Most of the OFF mutants showed a relatively slower rate in phosphorylation reception without much difference in the final outcome compared to WT, with the only exception of P106S that showed a 50% reduction in phosphotransfer within 30 min (Fig 7A–7D). However, the effects of these mutations on protein-DNA-binding were strikingly disruptive (Fig 7E), further confirming that even a small difference in the kinetic rate can lead to severe defects in protein function, consistent with phosphotransfer being the rate limiting step for signal transduction. As anticipated, all ON mutants exhibited much higher affinity to the target promoter. M58I mutation facilitates phosphotransfer from HitS to HitR, which in turn promotes DNA-binding with a 5-fold increase in binding affinity. Both ON mutations (K168A and E203A) from the DNA-binding domain had no significant effects on phosphotransfer kinetics. However, these two mutations dramatically enhanced protein-DNA-binding with 10-fold higher affinity compared to that of WT (Fig 7E–7G). We hypothesized that these two ON mutants might bypass phosphorylation and bind DNA with greater affinity without phosphorylation-mediated activation.

To test the essentiality of phosphorylation-triggered activation for HitR-DNA-binding, we repeated the EMSA experiments using WT HitR and the three ON mutants in the absence of phosphorylation. HitR WT or mutant were incubated directly with radioactively labelled DNA probe and the binding affinity was examined. Surprisingly, HitR WT binds to DNA with comparable affinity regardless of phosphorylation activation (S6 Fig), likely due to overexpression of HitR in *E. coli* that led to a conformational transition from an inactive to active-like state as observed for KdpE previously [43]. However, in the absence of phosphorylation, the DNA-

binding activity of the M58I mutant was completely abolished with up to 800 nM of the mutant protein tested. K168A showed similar DNA-binding activity as WT while E203A only preserved minimal activity (S6 Fig), which could be explained by the influence of these mutations on HitR dimerization status: K168A facilitated dimerization while E203A disrupted dimerization *in vitro* (Fig 3F), signifying the importance of dimerization in HitR activation. Furthermore, it is clear that phosphorylation-triggered activation is crucial for HitR-DNA-binding. Taken together, we conclude that the receiver and DNA-binding domains communicate through critical residues as they work together to ensure HitR phosphorylation and downstream gene activation in response to specific stimuli.

## Residues critical for HitS-HitR interaction

It is noteworthy that the HitS ON mutation of F149 to Leu in the RR-binding interface led to potent activation of HitRS signaling (S1A, S4E and S6A Figs and S3 Table). This is a typical functional trade-off mutation that affects all three activities with diminished autophosphorylation, enhanced phosphotransfer, and disrupted dephosphorylation (S3 Table), underscoring the significance of the RR binding interface in all three catalytic reactions of HitS. We reasoned that this mutant, along with other mutants from Helix 1, would affect HitS-HitR interaction. To test this idea, we determined HitS-HitR binding affinity *in vitro* using microscale thermophoresis. The two WT proteins bound to each other with an affinity of 385nM (S7A Fig), and activated HitS modestly enhanced the binding affinity with a $K_d$ value of 270 nM (S7B Fig). The difference was not dramatic but could be physiologically relevant during HitRS signaling *in vivo* since some RRs exhibit a reduced affinity for their cognate kinases upon phosphorylation [44]. Surprisingly, both HitR$^{S141L}$ and HitR$^{F149L}$ drastically disrupted the protein-protein interaction. Specifically, when the mutants were not activated, the binding affinity was 4–6 times lower compared to WT (S7C and S7E Fig). When HK is not activated through autophosphorylation, it is in phosphatase-competent state. These data explained why the phosphatase activity of these two mutants (HitR$^{S141L}$ and HitR$^{F149L}$) was disrupted (Fig 6A and 6D). By contrast, activation of these two mutants mediated by autophosphorylation improved their binding affinity to HitR, although still significantly weaker than WT (S7D and S7F Fig). These data indicate that these two residues (S141 and F149) of HitS are important for HK-RR interaction particularly during dephosphorylation.

## Discussion

Recent structural and biochemical work has provided valuable signaling models and substantially deepened our understanding of the molecular basis of signal transduction from external input domains to cytoplasmic output domain. Heretofore, there are more than 600 three-dimensional structures of HKs and RRs available; however, most of these structures are for individual domains, particularly for the membrane-bound HKs. It is challenging to obtain high-resolution structures of full-length proteins due to solubility, flexibility, and dynamics of the sensor HKs. Individual domains have inherent features and functional modes, but their interactions with other partners are crucial for specific signaling pathways and regulatory mechanisms. In this study, robust and unbiased genetic selections enabled isolation of point mutations within HitRS that constitutively switch on or off signal transduction of this TCS and these point mutations were further characterized systematically. These data demonstrated the effects of these mutations on diverse activities intrinsic to TCS signaling, defined the critical residues that are involved in HitRS signal transduction, determined phosphotransfer as the rate-limiting step for signal transduction, and shed light on the signaling mechanism of each individual domain and the TCS as a whole.

## Hydrophobic interactions within HAMP domain

HAMP domains in HKs function as signal transducing modules that couple conformational changes of sensory domains to the catalytic activity of the kinase core domains [15, 18]. HAMP domains can be swapped among proteins without compromised functionality [45, 46], indicative of a conserved signaling mechanism. A few models have been proposed for the mechanism of signal transduction through the HAMP domain [33, 35, 37, 38, 47, 48]. These models differ in many aspects but share one commonality: the HAMP domain shuttles between two distinct conformations, which represent two opposing signaling states and are stabilized by different subsets of conserved residues [18]. Indeed, we isolated a total of 17 point mutations from genetic selections, either constitutively activating ON (3) or inactivating OFF mutations (14), within this 50-residue HAMP domain (Fig 2). Each individual point mutation induces conformational changes sufficient for switching the signaling function to either an on or off state, which exemplifies the dynamics, flexibility, and interchangeable nature of the HAMP domain. Five hydrophobic residues identified are located at the dimer interface (I85, I88, A89, M117, and L121) and any mutations to neutral, hydrophilic, or even slightly less hydrophobic residues would drastically affect conformation of this domain resulting in loss of function (Fig 2). Thus, it is clear that these hydrophobic residues pack together in the interior of the helix bundle and stabilize protein conformation through hydrophobic interactions. Furthermore, among three constitutively ON mutations isolated from the HAMP domain, two were substitutions from neutral residues (P72 and T118) to hydrophobic residues (Leu and Ile, respectively; Fig 2). Both Leu and Ile are highly hydrophobic and custom-made for introducing additional hydrophobic effects to stabilize the kinase-competent conformation of this helix bundle. In conclusion, this and other studies demonstrated the importance of hydrophobic residues in HAMP function and the essentiality of this domain in TCS signaling.

## The kinase core: DHp and CA domains and their interaction

The DHp domain forms a homodimeric antiparallel four-helix bundle with two α-helices joined by a hairpin loop (S1A and S1B Fig). Three catalytic reactions take place at this domain: (i) histidine autophosphorylation, (ii) phosphotransfer from HK to its cognate RR, and (iii) dephosphorylation of the phosphorylated RR. This helix bundle can be divided into three segments based on prior DHp sequence analysis [15] (S1B Fig). Below we summarize the role for each segment and the effects of mutations in that segment have on the function.

First, the top region serves as the binding site for the Gripper fingers of CA during autophosphorylation. It switches between symmetric and asymmetric conformations, which correlate with phosphatase-competent and kinase-competent states, respectively [15]. Five ON mutations were isolated from this region (S136L, L184F, L185R, T188I, and L189P) (S1B Fig). The sidechain of S136 or T188 likely forms a hydrogen bond with the protein backbone and its substitution to a hydrophobic residue (Leu or Ile) introduces hydrophobic interactions within the four-helix bundle and enables HAMP helices to shift conformation towards a kinase-competent state. On the other hand, the hydrophobic Leu (L184, L185, or L189) was mutated to a relatively less hydrophobic (L184F), hydrophilic (L185R), or sterically restricted residue (L189P) (S1B Fig), all of which resulted in asymmetric kinase-on conformation. Second, the highly symmetric central core follows immediately after the well-conserved phosphoaccepting His and functions as the docking site for the cognate RR during either phosphotransfer or dephosphorylation [15]. The conserved (T/N)-P dipeptide is known to form a kink for helix bending that allows the N-terminus of the HAMP1 helix to adopt multiple conformations during signaling [15]. HitS contains a Ser, the preferred substitution residue for Thr, along with a conserved Pro at this position. Either residue could be mutated (S141L or P142S) to disrupt this

tight turn on the protein surface and lock the kinase in a constitutively on conformation (S1B Fig). In addition, L177 located in the interior of the four-helix bundle was mutated to a much less hydrophobic Ala and this mutation also triggered sufficient conformation changes from stable symmetric to asymmetric state, indicating the involvement of L177 in the dimer interface. Furthermore, both S141L and F149L mutations disrupted the interaction of HitS with its cognate regulator HitR particularly during dephosphorylation (S7C and S7E Fig) resulting in potent activation of HitRS signaling (S3D and S3E Fig). These data illustrate the importance of this segment for HK-RR interactions and TCS signal transduction. Third, the bottom part of the bundle is the continuity of the RR-binding interface joined by a hairpin loop. The sequence and length of this region are highly variable (S1A Fig), reflecting sequence-specific interactions for recognition of the cognate RR. In conclusion, these results strongly support prior DHp sequence analysis [49] and demonstrate that the symmetry-asymmetry transition is a key feature of signal transduction through the DHp domain.

The CA domain is well conserved with N, G1 (or D), F, G2, G3 sequence motifs (S1A and S1C Fig), which are all defined by the critical residues within these boxes and are all involved in ATP-binding and catalysis. In between the F and G2 boxes, a flexible loop called the ATP-lid covers the ATP-binding site (S1A and S1C Fig). The flexibility of the ATP-lid is important for ATP-binding as well as interacting with the DHp domain, which allows the CA domain to bind to different regions of DHp during multiple catalytic reactions depending on DHp conformation and the catalytic status of the CA domain [50–53]. A Gripper helix with four hydrophobic residues named Gripper fingers, located within the G2-box (S1A and S1C Fig), was recently defined and works together with a Phe in the F-box to mediate the interaction with the DHp domain [15]. Interestingly, most of the point mutants isolated within the CA domain were located in this region: four in the ATP-lid (A300E, N305K, R306S, and G309R) and two within the Gripper helix (A316E and K320E) (S1A and S1C Fig). Each individual mutation triggers structural changes of the CA domain and the interacting partners of CA and shifts the conformation equilibrium of the entire HK towards kinase-on state, highlighting the flexibility and versatility of these two motifs. Three of these mutants were characterized *in vitro*: both A300E and R306S promoted autokinase and phosphotransfer activities while G309R enhanced phosphotransfer with drastically diminished autophosphorylation. In addition, both A300E and G309R had no impact on phosphatase activity while R306S significantly disrupted dephosphorylation of the phosphorylated HitR (S3 Table), demonstrating the involvement of CA domain in all three catalytic activities. In conclusion, these data along with other structural analyses [15, 16, 18] support a model in which the CA domain needs to adopt several positions relative to the DHp domain during different catalytic states of the HK, and the interaction between DHp and CA domains mediated by the Gripper helix are critical for TCS signaling.

## The receiver and DNA-binding domains of RR and regulation mechanism

The receiver domains share a conserved (βα)5 fold (S2A and S2C Fig) and function as phosphorylation-dependent switches to modulate the activity of the effector domain using distinct inter- and/or intramolecular interactions in the inactive and active states [54]. Some RRs have been reported to dimerize in two different orientations: one involves the α4-β5-α5 surface and the other involves the α1/α5 surface and only the α4-β5-α5 dimer is functional upon activation [40]. The majority of the exposed sidechains on the α4-β5-α5 surface are hydrophilic, suggesting that any interactions through this interface are likely to be dynamic [40, 55]. Indeed, two residues (F95 and P106) identified from the *relE* selection are located in this region: both mutants (F95S and P106S) were still capable of accepting phosphor signals from HitS with rather slower kinetic rates but the DNA-binding ability of both mutants was nearly abolished

(Fig 7 and S3 Table), indicating the dimerization step between phosphotransfer and DNA-binding is likely disrupted. The hydroxyl group in the sidechain of Ser is fairly reactive and may form hydrogen bonds with the polar residues in the α4/α5 helices and disrupt the dynamics and flexibility of this α4-β5-α5 dimer interface (S2C Fig), which in turn prevents HitR from dimerizing through this interface thereby abrogating HitR activation (S4 and S5 Figs). These data suggest the dynamics and flexibility of this α4-β5-α5 interface may be important for RR activation through dimerization in the correct orientation. A conserved Met (M58) at two residues after β3 strand (S2A and S2C Fig) was identified from the *ermC* selection and its mutation to a bulky and mostly hydrophobic Ile promoted the phosphoryl group transfer from His to Asp (Fig 7). It is unclear how Ile substitution at this position facilitates phosphotransfer. There are two possibilities: (i) introducing hydrophobic residues to protein surfaces can stabilize a protein through improved water–protein interactions as reported recently [56], or (ii) Ile is in close proximity to the dimer interface and can facilitate dimerization through hydrophobic effects. Nonetheless, M58I substitution led to conformational changes of the receiver domain resulting in activation of the associated DNA-binding domain and downstream gene transcription (S3K and S5 Figs).

RRs with DNA-binding domains can be categorized into four subfamilies: OmpR/PhoB, NarL/FixJ, NtrC/DctD, and LytR/AgrA. HitR belongs to the largest OmpR/PhoB subfamily with a conserved winged helix-turn-helix fold (S2D Fig). Helix α8 and the last two β-strands (β9/β10) are critical for DNA-binding: α8 helix recognizes the specific DNA sequence and inserts into the major groove of DNA while the β-hairpin (β9/β10) binds in the minor DNA groove [57]. Two residues were identified within the α8 helix by the genetic selections: R192 and E203 and one residue within strand β10: Y222. The sidechains of both polar residues (R192 and E203) likely participate in hydrogen bonding with specific bases in the major DNA groove. Interestingly, substitution of R192 to a hydrophobic Cys led to complete loss of DNA-binding while mutation of E203 to a hydrophobic Ala enhanced DNA-binding with a 10-fold increase in binding affinity compared to WT (Fig 7). Hydrophobic residues (Cys and Ala) can also interact with DNA bases and stabilize protein-DNA complexes through hydrophobic interactions although the completely opposite effects of these two mutations seemed counterintuitive. However, R192C disrupted phosphotransfer while E203A showed no evident effects on phosphotransfer, indicating that R192 but not E203 in helix α8 of the DNA-binding domain may interact with the receiver domain and facilitate transfer of the phosphoryl group from HK to RR. R192C mutation disrupted this interaction leading to diminished phosphotransfer and abolished DNA-binding while E203A enhanced DNA-binding likely through hydrophobic effects. Y222 is located only six residues away from the C-terminus of HitR, however, its substitution to acidic Asp drastically disrupted the β-strand conformation and led to protein instability and complete loss of DNA-binding (Figs 3B and 7E and S3 Table). This mutant was still able to receive phosphoryl group from HK with a lower kinetic rate (Fig 7A–7D), indicating that the β-hairpin may not be involved in phosphotransfer but plays a critical role in DNA-binding.

Collectively, this study provides a detailed characterization and structure-function analysis of an entire TCS, defines molecular determinants of each domain for both HK and RR, reveals residues critical for various activities intrinsic to TCSs, uncovers interaction specificity among different domains and between the HK and RR, and extends our understanding of the molecular basis of signal transduction through TCSs. In addition, all constitutively ON point mutants identified within these two well-conserved signaling proteins could be useful for studying fundamental mechanisms of signaling in other TCSs with unknown targets or unknown partners. These ON mutations could also serve as a blueprint for developing biotechnology tools suitable for synthetic biology engineering that connects sensory modules with signaling outputs [58].

Given that antibiotic resistance is one of the most significant threats to global health, novel antimicrobial therapeutics are in desperate need. It is reasonable to think that the well conserved HAMP domain found in many sensor and chemotaxis proteins could be the top target for rational inhibitor design to disrupt the hydrophobic interactions within the four-helix bundle and disrupt the signal transmission that is required for many biological processes. This study along with other findings pave the way for developing novel antimicrobials or adjunctive treatments that target signal transduction in infectious pathogens.

## Materials and methods

### Bacterial strains and growth conditions

All strains and plasmids used in this study are listed in S1 Table. Cells were grown in LB with vigorous shaking or on solid LB agar plates with appropriate antibiotic selection at 37˚C. The concentrations of antibiotics used are: carbenicillin (carb, 50 μg ml$^{-1}$), chloramphenicol (cam, 30 μg ml$^{-1}$), kanamycin (kan, 40 μg ml$^{-1}$), and erythromycin (erm, 20 μg ml$^{-1}$). Stocks of the compound '205 (50 mM) were made in DMSO and stored at -20˚C.

### DNA manipulation and strain construction

All strains used in this study were verified by PCR using primers listed in S2 Table. Electroporation of plasmids into *B. anthracis* and subsequent genetic manipulation were performed as previously described [12] with details as follows. The generation of plasmids for insertion of gene fusions at specific chromosomal loci was performed by inserting flanking sequences in the mutagenesis plasmid pLM4 [59]. Briefly, flanking sequences were amplified using a distal primer containing a restriction enzyme site found in pLM4 (XmaI or SacI) and a proximal primer containing a short sequence overlapping the adjacent flanking region and containing sites for the restriction enzymes NheI and SacI. PCR-amplified DNA was fused using PCR sequence overlap extension (PCR-SOE) as described previously [60] and inserted between the XmaI and SacI sites in pLM4. To construct a plasmid for integration into the pseudogene locus *bas3009*, the plasmid pLM4-*3009* was constructed as described above using primers *3009*_XmaI_fwd, *3009*_SOE-L, *3009*_SOE-R, and *3009*_SacI_rev. To construct a second plasmid for integration into the pseudogene locus *bas4599*, the plasmid pLM4-*4599* was constructed as described above using primers *4599*_XmaI_fwd, *4599*_SOE-L, *4599*_SOE-R, and *4599*_SacI_rev. To construct a third plasmid for integration into the pseudogene locus *bas4927*, the plasmid pLM4-*4927* was constructed as described above using primers *4927*_XmaI_fwd, *4927*_SOE-L, *4927*_SOE-R, and *4927*_SacI_rev.

To generate a plasmid for chromosomal integration of a *hit* promoter-*ermC* fusion, the *hitPRS* promoter was amplified from *B. anthracis* chromosomal DNA using primers $P_{hit}ermC$_NheI_fwd and $P_{hit}ermC$_SOE_L; *ermC* was amplified from plasmid pCR2.1.*ermC* using primers $P_{hit}ermC$_SOE_R and $P_{hit}ermC$_KpnI_rev. PCR amplified DNA was fused using PCR-SOE and inserted between the NheI and KpnI sites of pLM4-*3009* to generate pLM4-*3009*::$P_{hit}ermC$. To construct plasmids for chromosomal integration of *hit* promoter-*relE* fusions, the *hitPRS* promoter was amplified from *B. anthracis* chromosomal DNA using primers $P_{hit}relE$_NheI_fwd and $P_{hit}relE$_SOE_L; *relE* was amplified from *E. coli* DH5α chromosomal DNA using primers $P_{hit}relE$_SOE_R and $P_{hit}relE$_KpnI_rev. PCR amplified DNA was fused using PCR-SOE and inserted between the NheI and KpnI sites of pLM4-*3009* and pLM4-*4599* to generate pLM4-*3009*::$P_{hit}relE$ and pLM4-*4599*::$P_{hit}relE$, respectively. To construct a plasmid for chromosomal integration of a supplemental copy of *hitRS*, *hitRS* was amplified from *B. anthracis* chromosomal DNA using primers *hitRS*_KpnI_fwd and

*hitRS*_NheI_rev and PCR amplified DNA was inserted between the NheI and KpnI sites of pLM4-*4927* to generate pLM4-*4927*::*hitRS*.

To construct plasmids for chromosomal *hitRS* point mutations, *hitRS* and flanking DNA was amplified from *B. anthracis* chromosomal DNA using primers *hitRS*_XmaI_fwd and *hitRS*_SacI_rev. PCR amplified DNA was inserted between the XmaI and SacI sites of pLM4 to generate pLM4-*hitRS*. Point mutations in *hitRS* were introduced using Q5 site-directed mutagenesis (NEB) following the manufacturer's recommendations and were confirmed by Sanger sequencing. In all cases, single or multiple rounds of mutagenesis were performed using these vector as described previously [59] and confirmed by PCR and Sanger sequencing.

To construct the point mutants of HitR or HitS, the *hitR or hitS* genes were amplified by PCR from the *B. anthracis* genome and cloned into pET15b expression vector to generate a 6x-histidine-tagged protein at the N-terminus. Point mutations in these two genes were introduced using site-directed mutagenesis. The recombinant plasmids were transformed into *E. coli* DH5α, confirmed by Sanger sequencing, then subsequently transformed into *E. coli* BL21 (DE3) pREL for protein expression and purification.

## Genetic selections

To select for inactivating mutations in *hitRS* or other genes required for HitRS function, we utilized strains *bas3009*::*hit-relE bas4599*::*hit-relE* (referred to herein as 2x(*relE*)) and *bas3009*::*hit-relE bas4599*::*hit-relE bas4927*::*hitRS* (referred to herein as 2x(*relE + hitRS*)). Strains were grown in 3 ml LB for 18 hours at 30˚C with shaking at 200 rpm. Next, dilutions in sterile deionized water were assembled (typically 1:10, 1:30, 1:100, and 1:300 for 2x(*relE*) and 1:1, 1:3, 1:10, and 1:30 for 2x(*relE + hitRS*)). 100 μl of each dilution was plated onto LB with 2 μM '205 and LB containing 4 μM '205 and plates were incubated at 30˚C for up to 5 days. To select for constitutively activating mutations in *hitRS* or mutations in other genes that suppress HitRS function, we utilized strain *bas3009*::*hit-ermC* (referred to herein as P$_{hit}$*ermC*). Selections were performed as described above, except that dilutions were typically 1:3, 1:10, 1:30 and 1:100 and dilutions were plated onto LB containing 5, 10, or 20 μg ml$^{-1}$ erythromycin. In all cases, spontaneously arising colonies exhibiting decreased '205 sensitivity or constitutive erythromycin resistance were selected, streaked for single colonies on fresh LB, and saved for further analysis.

## Growth curves

Cells were grown overnight in LB medium at 30˚C, subcultured at a 1:100 ratio into fresh LB medium and grown for 6 h at 37˚C with vigorous shaking. Cell density (OD$_{600}$) was monitored every 30 min for 24 h at 37˚C with continuous shaking using a BioTek Epoch2 spectrophotometer. The HitRS inducer '205 (20 μM) and/or 20 μg ml$^{-1}$ erythromycin were used as described. Experiments were conducted at least three times with three biological replicates each time. Data shown are averages of three replicates (mean ± STD).

## RNA extraction and quantitative real-time PCR (qRT-PCR)

Cells were grown at 30˚C in LB medium overnight and subcultured at a 1:100 ratio into fresh LB medium. After 6 h of growth at 37˚C in the presence or absence of 20 μM '205, aliquots of 4 ml of cell culture were harvested by centrifugation. Total RNA was extracted using RNeasy Mini Kit following the manufacturer's instructions (Qiagen Sciences, Germantown, MD), and treated with Turbo-DNA free DNase (Ambion). The DNase was removed using DNase removal reagents (Ambion). RNA samples were quantified using a NanoDrop spectrophotometer. Two hundred nanograms of total RNA from each sample was subjected to cDNA

synthesis using high-capacity cDNA reverse transcription kits (Applied Biosystems, Foster City, CA). Quantitative PCR (qPCR) was then conducted using iQ SYBR green supermix (Bio-Rad) on a CFX96 qPCR cycler (Bio-Rad). The *B. anthracis* housekeeping gene 16S rRNA was used as an internal control.

### Homology modeling

Homology models of the HAMP (residue 66 to 121) and intracellular (residue 113 to 352) domains of HitS and full-length protein of HitR were generated using I-TASSER [32] or Phyre2 [39] with default settings. The respective homology models with the highest confidence and coverage were employed. Online programs including SMART [26], MemBrain [61], and TOPCONS [27] were used to predict the modular structure arrangement of these two proteins.

### Protein expression and purification

All proteins (WT and mutants) were expressed in *E. coli* BL21 (DE3) pREL. Cells were grown overnight in LB medium at 30°C, subcultured at a 1:100 ratio into fresh LB medium and grown for ~4 h at 30°C with vigorous shaking ($OD_{600}$~0.6–0.8). Cell cultures were then switched to incubation at 18°C and IPTG (0.5 mM) was added to induce protein expression. After 18 h of incubation at 18°C, cells were harvested by centrifugation. Cells were lysed by sonication and the lysate was centrifuged at 20,000 x g for 45 min at 4°C to remove the debris. The subsequent clarified lysate was loaded onto an NTA-Ni column, and protein was eluted sequentially with increasing concentration of imidazole. The fractions containing high-purity proteins were pooled and subjected to dialysis for buffer exchange using storage buffer containing 50 mM Tris (pH 8), 300 mM NaCl, 0.5 mM EDTA, 1 mM DTT, and 20% glycerol, and concentrated using Amicon concentrators. Purified proteins were subjected to SDS-PAGE for quality control analysis and quantified using absorbance at 280nm on a NanoDrop spectrophotometer. Aliquots of purified proteins were flash-frozen in liquid nitrogen and stored at -80°C.

### Size exclusion chromatography

Size exclusion chromatography was carried out using Superdex 75 prep grade resin equilibrated in SEC buffer (20 mM Tris pH 8.0, 300 mM NaCl, 0.1 mM EDTA, 10% glycerol, 0.5 mM TCEP) with a flow rate of 1ml min$^{-1}$. The experiments were performed in triplicates. Each protein (WT or mutant) was injected into the column individually at a final concentration of 0.5 mg ml$^{-1}$. The absorbance at 280 nm was recorded and data were analyzed using GraphPad Prism 8.

### Autophosphorylation assay

To evaluate the effects of mutations on HitS autokinase activity, the autophosphorylation efficiency of HitS WT and mutants was investigated. The autophosphorylation reactions were assembled as follows: 5 μM HitS WT or mutant, 1x kinase buffer (50 mM Tris pH 8.0, 10 mM $MgCl_2$, 100 mM KCl, 1 mM dithiothreitol, and 10% glycerol), 20 μM ATP, and 5 μCi of [γ-$^{32}$P]-ATP. The reactions were started with addition of the [γ-$^{32}$P]-ATP and incubated at 37°C for 30 min. SDS loading buffer (50 mM Tris-Cl pH 6.8, 2% (w/v) SDS, 0.1% (w/v) bromophenol blue, 10% (v/v) glycerol, and 5% 2-mercaptoethanol) was added into each reaction mixture to stop autophosphorylation. The resultant mixtures were subject to SDS-PAGE. The same amount of protein used for each reaction was subject to SDS-PAGE and served as quantity control. The experiments were carried out four times.

To understand how these ON mutations affect the kinetics of autophosphorylation, a time-course study was performed for HitS WT and mutants, in which the autophosphorylation reaction was monitored for 15 min at 37°C. Aliquots of reaction mixtures were collected, mixed with SDS loading buffer, and kept in ice to stop autophosphorylation. The resultant mixtures were subject to SDS-PAGE. The experiments were performed three times for each protein tested. After electrophoresis, the gels from both sets of autophosphorylation experiments were dried using a gel dryer, exposed to a phosphorimager screen overnight, and scanned by a phosphor image analyzer (Typhoon FLA 7000). The intensity of the phosphor-signal was quantified using GelQuantNET. The first four timepoints (0, 1, 2, and 3 min) from the kinetic study were used for slope determination by linear regression analysis using Graph-Pad Prism 8.

## Phosphotransfer assay

Three sets of phosphotransfer experiments were carried out as follows. (i) To evaluate the effects of HitS mutations on the kinetics of phosphor-signal transferring, phosphor-transferring efficiency of HitS WT and ON mutants to HitR WT was examined. HitS WT or mutant was autophosphorylated with $[\gamma\text{-}^{32}P]$-ATP as described above in an 80 μl reaction volume for 30 min, 10 μl of which was then sampled into SDS-PAGE loading buffer, kept on ice, and served as time zero. To the remaining 70 μl of reaction mixture, HitR WT (5 μM of final concentration) was then added and phosphotransfer from HitS WT or mutant to HitR WT was then monitored for 15 min at room temperature. Aliquots of reaction mixtures were collected at different timepoints as indicated, amended with SDS-PAGE loading buffer, kept on ice, and subjected to SDS-PAGE.

(ii) To examine the effects of HitR mutations on signal reception, the efficiency of phosphotransfer from HitS WT to HitR WT or mutants was quantified. HitS WT was incubated with $[\gamma\text{-}^{32}P]$-ATP for 30 min at 37°C as described above and the phosphor signal was subsequently transferred to HitR WT or mutant with incubation for 30 min at 37°C. SDS-PAGE loading buffer was added into each reaction and the resultant mixtures were subject to SDS-PAGE. The same amount of protein used for each reaction was subject to SDS-PAGE and served as quantity control. Four independent experiments were carried out to determine the phosphotransfer efficiency.

(iii) To assess the effects of HitR mutations on the kinetics of phosphor signal transferring, phosphor-transferring efficiency of HitS WT to HitR WT or mutants was examined. HitS WT was autophosphorylated with $[\gamma\text{-}^{32}P]$-ATP as described above in a 100 μl reaction volume for 30 min, 10 μl of which was then sampled into SDS-PAGE loading buffer, kept on ice, and served as time zero. To the remaining 90 μl of reaction mixture, HitR WT or mutants (5 μM of final concentration for each protein) was then added and phosphotransfer from HitS WT to HitR WT or mutants was then monitored for 30 min at room temperature. Aliquots of reaction mixtures were collected at different timepoints as indicated, amended with SDS-PAGE loading buffer, kept on ice, and subjected to SDS-PAGE. The experiments were performed three times for each protein tested. After electrophoresis, the gels from all three sets of phosphotransfer experiments were dried using a gel dryer, exposed to a phosphorimager screen overnight, and scanned by a phosphor image analyzer (Typhoon FLA 7000). The intensity of radioactive signals was quantified accordingly using GelQuantNET.

## Dephosphorylation assay

To assess the impact of HitS mutations on its phosphatase activity, dephosphorylation efficiency of HitS WT or mutants was examined. The GST-PmrBc fusion protein [42] prepared

with glutathione sepharose beads served as a phosphor donor and was autophosphorylated by incubation with $[\gamma\text{-}^{32}P]$-ATP for 30 min at 37°C. The phosphorylation signal was subsequently transferred to HitR WT protein. The resultant phosphorylated HitR WT protein was isolated by centrifugation, purified using Micro Bio-Spin columns to remove any remaining free ATP, and the phosphatase activity of HitS WT or mutants was then monitored for 60 min in the same kinase buffer as described above. The intensity of phosphor signal at different timepoints was quantified. Data from three independent experiments for each protein (WT or mutant) were plotted and analyzed using GraphPad Prism 8.

## Electrophoretic mobility shift assay (EMSA)

The promoter region (202bp) of the *hitPRS* operon was amplified by PCR using a specific primer set listed in S2 Table. Two hundred nanograms of purified DNA was labelled at 5'-ends with $[\gamma\text{-}^{32}P]$-ATP using T4 polynucleotide kinase. After labelling, G10 columns (NucAway spin columns, Invitrogen) were used to remove the unincorporated $(\gamma\text{-}^{32}P)$ ATP and radioactivity of the probe was quantified using a scintillation counter. Two sets of EMSAs were conducted: (i) To examine the effects of mutations on HitR-DNA-binding, DNA-binding of HitR WT or ON mutants to the target promoter was evaluated in the absence of phosphorylation-mediated activation of the regulator; (ii) to check the DNA-binding ability of HitR WT or ON mutants in their phosphorylation-active form, autophosphorylation of HitS WT was carried out as described above for 30 min, HitR WT or mutant was added to the reaction mixture and incubated for 5 min at room temperature to allow sufficient phosphotransfer. The activated HitR WT or mutant was then used to set up EMSA binding assay as follows: ~ 1 fmol of labelled DNA probe, 100 mM NaCl, 0.5 µg poly(dIdC), varied concentration of HitR WT or mutant protein, and 1X binding buffer (10 mM Tris-HCl, pH 8.0, 5% glycerol, 1 mM EDTA, 1 mM DTT, 60 mM potassium glutamate, 150 µg ml$^{-1}$ BSA). The reactions were incubated at room temperature for 20 min and then subjected to electrophoresis in a 5% Mini-PROTEAN TBE Gel. After electrophoresis, the gels were dried using a gel dryer, exposed to a phosphorimager screen overnight, and scanned by a phosphor image analyzer (Typhoon FLA 7000). The band intensity of unbound DNA was quantified using GelQuantNET. All the data points from three independent experiments were plotted and subjected to $K_d$ determination using GraphPad Prism 8.

## Microscale thermophoresis assay (MST)

The His-tagged HitR WT protein was labelled with Monolith RED-tris-NTA fluorescence dye according to the manufacturer protocol (Cat# MO-L018, NanoTemper Technologies, Germany) in MST buffer (50mM HEPES pH8.0, 300mM NaCl, 0.1mM EDTA). The labeled HitR at a concentration of 50 nM was used for each reaction with fluorescence count signal of 250 units or greater. The ligand stock solutions (HitS WT or mutants) were serially diluted in 16 steps using MST buffer. In parallel, HitS WT or mutant protein was autophosphorylated at 37°C for 30 min before incubation with the labelled HitR. The protein-dye mixtures were incubated for 30 min in the dark at room temperature and then loaded into standard Monolith capillaries. The capillaries were subsequently scanned using a Monolith NT.115 MST instrument with setting at 40% MST and 40% excitation. All data points from each independent experiment were plotted and subjected to $K_d$ determination using the MO Affinity Analysis software. The $K_d$ values shown are average of three independent experiments (mean ± SE).

## Supporting information

**S1 Table. Bacterial strains and plasmids used in this study.**
(PDF)

**S2 Table. Oligonucleotide primers used for this study.**
(PDF)

**S3 Table. Point mutations within HitRS affect the signaling pathway in various manners.**
(PDF)

**S1 Fig. Conserved motifs in the kinase core domains.** (A) Multiple HK sequence were aligned to display the conserved motifs in the kinase core domains of HKs. The two helices of DHp domain are indicated and conserved hydrophobic residues in helix 2 are shaded in gray. The H, N, D (or G1), F, G2, and G3-boxes are shown in boxes. The RR-binding motif is underlined, the hydrophobic residues in the HAMP2 helix are highlighted in gray, and the hydrophobic residues that form the Gripper fingers are highlighted in cyan. The residues between F and G2-boxes form a loop structure known as ATP-lid that holds the ATP molecule. The ATP-lid, Gripper helix, N, D, and G-boxes are involved in ATP-binding and phosphotransfer. The residues identified from genetic selections are highlighted in either orange or blue to specify either ON or OFF mutations, respectively. The sequences are the following: HitS, *B. anthracis*; HK853, *Thermotoga maritima*; PhoQ, *E. coli*; EnvZ, *E. coli*; WalK, *Staphylococcus aureus*; PhoR, *E. coli*; ArcB, *E. coli*. (B) The four-helix bundle of DHp domain is divided into three segments based on sequence analysis [15] and all residues highlighted in orange or blue were identified by genetic selections. The homology modelling was based on HK853 (PDB ID: 4JAU). (C) All features in the CA domain are highlighted in the model structure, which was based on HK853 (PDB ID: 4RH8). OFF mutations are shown in blue while ON mutations are shown in orange.
(PDF)

**S2 Fig. The receiver and DNA-binding domains of RR.** (A) Sequence alignment of the receiver domains. The sequences are the following: HitR; BfmR, *Acinetobacter baumannii*; PhoP, *Bacillus subtilis*; YycF, *Bacillus subtilis*; PhoB, *E. coli*; and RstA, *A. baumannii*. (B) Sequence alignment of the DNA-binding domains. The sequences are the following: HitR; PrrA, *Mycobacterium tuberculosis*; PhoB, *E. coli*; SaeR, *Staphylococcus aureus*; VicR, *Streptococcus mutans*; RegX3, *Mycobacterium tuberculosis*; YycF, *Bacillus subtilis*. The α-helices and β-sheets are underlined. The homology models of the receiver (C) and DNA-binding domain (D) of HitR were generated based on *M. tuberculosis RegX3* (PDB ID: 2OQR), which is in an active dimer form. Residues identified from genetic selections are highlighted in either orange or blue to specify either ON or OFF mutations, respectively.
(PDF)

**S3 Fig. HitRS point mutants are resistance to erythromycin.** Growth of *B. anthracis* WT, WT $P_{hit}ermC$, and isolated activating suppressors in vehicle, erythromycin (20 μg ml$^{-1}$), or 20 μM '205 plus 20 μg ml$^{-1}$ erythromycin was monitored for 24 h. Data are averages of three independent experiments (mean ± SEM).
(PDF)

**S4 Fig. HitRS point mutants are resistant to '205-mediated killing.** Growth of *B. anthracis* WT, WT 2(*relE*) (A-F; WT *bas3009*::$P_{hit}relE$ *bas4599*::$P_{hit}relE$), WT 2(*relE+ hitRS*) (G-N; WT *bas3009*::$P_{hit}relE$ *bas4599*::$P_{hit}relE$ *bas4927*::*hitRS*), and isolated inactivating suppressors. Strains were grown in medium containing vehicle (A, C, E, G, I, K, M) or 20 μM '205 (B, D, F, H, J, L, N) and growth was monitored for 24 h. Data are averages of three independent experiments (mean ± SEM).
(PDF)

**S5 Fig. Point mutations within HitRS affect transcription of *hitPRS*.** To evaluate the effects of the point mutations on transcription of the *hitPRS* operon, qPCR was performed. Briefly, overnight cultures were inoculated with a 1:100 ratio into fresh LB medium without (A, C, E; vehicle) or with 20 μM '205 (B, D, F). After 6 h of vigorous shaking at 37°C, cells were harvested, and total RNA was extracted and subjected to cDNA synthesis followed by qPCR quantification. All point mutations selected for biochemical characterization were tested: (A-B) constitutively activating and (C-D) inactivating mutants. To confirm the results from the genetic selections using different constructs, representative point mutations were reconstructed in *B. anthracis* WT background and the effects of these mutations on transcription of the *hitPRS* operon were tested using qPCR without (E, vehicle) or with 20 μM '205 induction (F). The mRNA level of each gene tested in WT untreated cells is set as 1 in both panels. Two sets of oligonucleotide primers were designed to check *hitP* transcript (*hitP*1 and *hitP*2). *B. anthracis* 16S rRNA was used as a housekeeping control gene. The data are expressed as the mean ± SEM (n = 3).
(PDF)

**S6 Fig. Phosphorylation-mediated activation is crucial for HitR-DNA-binding.** To examine the effects of mutations on HitR-DNA-binding, DNA-binding of HitR WT or ON mutants to the target promoter was evaluated using electrophoretic mobility shift assay (EMSA). HitR WT or ON mutants were not subjected to phosphorylation-mediated activation prior to EMSA. The experiments were repeated three times and representative images are shown.
(PDF)

**S7 Fig. Residues critical for HitS-HitR interaction.** To evaluate the effects of mutation on HitS binding ability to its cognate regulator HitR, a microscale thermophoresis assay was carried out. Briefly, HitR WT protein was labelled with a RED-tris-NTA fluorescence dye at room temperature for 30 min, and 50 nM of the labelled HitR was subsequently incubated with varied concentrations of HitS WT (A) or mutant protein (C and E). In parallel, HitS WT (B) or mutant protein (D and F) was autophosphorylated at 37°C for 30 min before incubation with the labelled HitR. All data points from each independent experiment were plotted and subjected to $K_d$ determination. The $K_d$ values shown are average of three independent experiments (mean ± SEM).
(PDF)

## Acknowledgments

We thank Jocelyn Simpson for her help on *ermC* selection. We thank Dr. Heather Kroh for her technical advice and Dr. Maria Hadjifrangiskou for sharing the GST-PmrBc construct. We thank members of the Skaar Laboratory for critical comments of the manuscript.

## Author Contributions

**Conceptualization:** Hualiang Pi, Eric P. Skaar.

**Data curation:** Hualiang Pi, Michelle L. Chu, Samuel J. Ivan, Casey J. Latario, Allen M. Toth, Sophia M. Carlin, Gideon H. Hillebrand, Hannah K. Lin, Jared D. Reppart, Devin L. Stauff.

**Formal analysis:** Hualiang Pi.

**Funding acquisition:** Eric P. Skaar.

**Investigation:** Hualiang Pi.

**Methodology:** Hualiang Pi.

**Project administration:** Eric P. Skaar.

**Supervision:** Eric P. Skaar.

**Validation:** Hualiang Pi.

**Visualization:** Hualiang Pi.

**Writing – original draft:** Hualiang Pi.

**Writing – review & editing:** Hualiang Pi, Eric P. Skaar.

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
