## [Decision Letter · Decision Letter 0]

2 Nov 2020

Dear Dr. Skaar,

Thank you very much for submitting your manuscript "Directed evolution reveals the mechanism of HitRS signaling transduction in Bacillus anthracis" for consideration at PLOS Pathogens. As with all papers reviewed by the journal, your manuscript was reviewed by members of the editorial board and by several independent reviewers. The reviewers appreciated the attention to an important topic. Based on the reviews, we are likely to accept this manuscript for publication, providing that you modify the manuscript according to the review recommendations.

Sincerely,

Theresa M. Koehler

Associate Editor

PLOS Pathogens

Michael Wessels

Section Editor

PLOS Pathogens

Kasturi Haldar

Editor-in-Chief

PLOS Pathogens

orcid.org/0000-0001-5065-158X

Michael Malim

Editor-in-Chief

PLOS Pathogens

orcid.org/0000-0002-7699-2064

Reviewer Comments (if any, and for reference):

Reviewer's Responses to Questions

**Part I - Summary**

Reviewer #1: In this study, Pi et al use a clever genetic design to perform a functional analysis of HitSR, a two-component system (TCS) of Bacillus anthracis (Ba). They isolate constitutively active or inactive variants. They generate and purify representative recombinant variants using E. coli for biochemical characterization of autokinase, phosphatase, phosphotransfer and DNA binding activities. The authors identify residues essential for each activity and for interaction between subdomains as well as between HitS and HitR. Some of the results are “as expected”.

The design for selection of mutants can be generalized to the study of other TCS. Data and results are extensively explained, well-presented and conclusive. While this particular TCS belongs to a pathogen, the study is strictly limited to the structure-function analysis of the HitRS and does not investigate an impact on the pathogenesis of B. anthracis nor an impact on antibiotic resistance.

Reviewer #2: The manuscript by Hualiang et al. characterizes the HitRS two component system, which has been implicated in detecting cell envelope stress, in Bacillus anthracis. The authors devised two genetic selections performed in B. anthracis to identify either critical residues in HitRS or constitutively active forms of HitRS. This strategy revealed gain-of-function mutations in the sensor domain and TM region of HitS and critical residues in the HAMP domain of HitS. The authors also report the in vivo consequence for B. anthracis growth in cells harboring various isolated HitRS alleles, examine protein stability and dimerization potential of the various mutants, and identify residues that are critical for HitR-HitS interaction.

**Part II – Major Issues: Key Experiments Required for Acceptance**

Reviewer #1: l.87-88: the authors cite compound VU0120205 ‘205 as a cell-envelope acting compound but the associated reference #23 describes compound VU0038882 (‘882) that activates HssRS by inducing endogenous heme biosynthesis in S. aureus.

I wonder if the authors could clarify the relationship between HssRS and HitRS. I am not sure how responding to heme availability (HssRS) fits with sensing cell envelope perturbations (HitRS). Also, the authors should explain why they picked HitRS over HssRS.

Reviewer #2: (No Response)

**Part III – Minor Issues: Editorial and Data Presentation Modifications**

Reviewer #1: l. 28: why is Ba referred as an intracellular pathogen?

l. 36: this is a very strong statement. I don’t think that the study comes close to laying such a foundation.

The “Author Summary” section uses slightly different words but essentially repeats the abstract.

l.49: it is odd to start the paper with a statement about the rising incidence of antibiotic resistance. That is not a trait associated with Ba. Word on the street is that this pathogen is dormant in soils; mostly! Certainly, Ba isn’t roaming around hospitals with the Enterococci and the likes of them.

l.58: during infection Ba is only transiently intracellular.

Fig. 1 and corresponding Result section present clever approaches to isolate hit alleles with constitutive activity and mutations that inactivate signaling or signal transduction. However, this does not bring the authors closer to identifying the signal that activates HitRS. As expected mutations arising from these selections bypass the requirement for the true ligand of HitRS.

Reviewer #2: In general, the experiments were carefully controlled, and the manuscript was well-written so that it should be accessible to non-experts. This was a straightforward but exhaustive characterization of a two-component system in an important pathogen. As such, it should be of interest to the broad signal transduction community and to those who study bacterial pathogenesis. I have no major scientific comments and only one minor cosmetic comment that the authors may or may not wish to incorporate.

Minor comment:

1. Fig. S3. This schematic represents a major set of results of this study. Consider moving it to the main text (perhaps in Fig. 1?).

PLOS authors have the option to publish the peer review history of their article (what does this mean?). If published, this will include your full peer review and any attached files.

Reviewer #1: No

Reviewer #2: No
---

## [Editor Report · Decision Letter 1]

11 Nov 2020

Dear Dr. Skaar,

We are pleased to inform you that your manuscript 'Directed evolution reveals the mechanism of HitRS signaling transduction in Bacillus anthracis' has been provisionally accepted for publication in PLOS Pathogens.

Best regards,

Theresa M. Koehler

Associate Editor

PLOS Pathogens

Michael Wessels

Section Editor

PLOS Pathogens

Kasturi Haldar

Editor-in-Chief

PLOS Pathogens

orcid.org/0000-0001-5065-158X

Michael Malim

Editor-in-Chief

PLOS Pathogens

orcid.org/0000-0002-7699-2064
---

## [Editor Report · Acceptance letter]

9 Dec 2020

Dear Dr. Skaar,

We are delighted to inform you that your manuscript, "Directed evolution reveals the mechanism of HitRS signaling transduction in *Bacillus anthracis*," has been formally accepted for publication in PLOS Pathogens.

Best regards,

Kasturi Haldar

Editor-in-Chief

PLOS Pathogens

orcid.org/0000-0001-5065-158X

Michael Malim

Editor-in-Chief

PLOS Pathogens

orcid.org/0000-0002-7699-2064